# Learning to Guide and to Be Guided in the Architect-Builder Problem

**Paul Barde**[*][†]
Québec AI institute (Mila)
McGill University
bardepau@mila.quebec

**Tristan Karch**[*]
Inria - Flowers team
Université de Bordeaux
tristan.karch@inria.fr

**Derek Nowrouzezahrai**
Québec AI institute (Mila)
McGill University

**Clément Moulin-Frier**
Inria - Flowers team
Université de Bordeaux
ENSTA ParisTech

**Christopher Pal**[‡]
Québec AI institute (Mila)
Polythechnique Montréal
ServiceNow - Element AI

**Pierre-Yves Oudeyer**
Inria - Flowers team
Univ. Bordeaux
Microsoft Research Montreal

## Abstract

We are interested in interactive agents that learn to coordinate, namely, a *builder* – which performs actions but ignores the goal of the task, i.e. has no access to rewards – and an *architect* which guides the builder towards the goal of the task. We define and explore a formal setting where artificial agents are equipped with mechanisms that allow them to simultaneously learn a task while at the same time evolving a shared communication protocol. Ideally, such learning should only rely on high-level communication priors and be able to handle a large variety of tasks and meanings while deriving communication protocols that can be reused across tasks. The field of Experimental Semiotics has shown the extent of human proficiency at learning from a priori unknown instructions meanings. Therefore, we take inspiration from it and present the Architect-Builder Problem (ABP): an asymmetrical setting in which an architect must learn to guide a builder towards constructing a specific structure. The architect knows the target structure but cannot act in the environment and can only send arbitrary messages to the builder. The builder on the other hand can act in the environment, but receives no rewards nor has any knowledge about the task, and must learn to solve it relying only on the messages sent by the architect. Crucially, the meaning of messages is initially not defined nor shared between the agents but must be negotiated throughout learning. Under these constraints, we propose Architect-Builder Iterated Guiding (ABIG), a solution to the Architect-Builder Problem where the architect leverages a learned model of the builder to guide it while the builder uses self-imitation learning to reinforce its guided behavior. To palliate to the non-stationarity induced by the two agents concurrently learning, ABIG structures the sequence of interactions between the agents into interaction frames. We analyze the key learning mechanisms of ABIG and test it in a 2-dimensional instantiation of the ABP where tasks involve grasping cubes, placing them at a given location, or building various shapes. In this environment, ABIG results in a low-level, high-frequency, guiding communication protocol that not only enables an architect-builder pair to solve the task at hand, but that can also generalize to unseen tasks.

## 1 Introduction

Humans are notoriously successful at teaching – and learning from – each others. This enables skills and knowledge to be shared and passed along generations, being progressively refined towards mankind's current state of proficiency. People can teach and be taught in situations where there is no shared language and very little common ground, such as a parent teaching a baby how to stack blocks during play. Experimental Semiotics (Galantucci & Garrod, 2011), a line of work that

---

[*]Equal contribution.      [†]Work conducted while at Inria.      [‡]Canada CIFAR AI Chair.

studies the forms of communication that people develop when they cannot use pre-established ones, reveals that humans can even teach and learn without direct reinforcement signal, demonstrations or a shared communication protocol. Vollmer et al. (2014) for example investigate a co-construction (CoCo) game experiment where an architect must rely only on arbitrary instructions to guide a builder toward constructing a structure. In this experiment, both the task of building the structure and the meanings of the instructions – through which the architect guides the builder – are simultaneously learned throughout interactions. Such flexible teaching – and learning – capabilities are essential to autonomous artificial agents if they are to master an increasing number of skills without extensive human supervision. As a first step toward this research direction, we draw inspiration from the CoCo game and propose the *Architect-Builder Problem* (ABP): an interactive learning setting that models agents' interactions with *Markov Decision Processes* (Puterman, 2014) (MDPs). In the ABP learning has to occur in a social context through observations and communication, in the absence of direct imitation or reinforcement (Bandura & Walters, 1977). Specifically, the constraints of the ABP are: (1) the builder has absolutely no knowledge about the task at hand (no reward and no prior on the set of possible tasks), (2) the architect can only interact with the builder through communication signals (cannot interact with the environment or provide demonstrations), and (3) the communication signals have no pre-defined meanings (nor belong to a set of known possible meanings). (1) sets this work apart from Reinforcement Learning (RL) and even Multi-Agent RL (MARL) where explicit rewards are available to all agents. (2) implies the absence of tele-operation or third-person demonstrations and thus distinguishes the ABP from Imitation and Inverse Reinforcement Learning (IRL). Finally, (3) prevents the architect from relying on a fixed communication protocol since the meanings of instructions must be negotiated.

These constraints make ABP an appealing setting to investigate *Human-Robot Interaction* (HRI) (Goodrich & Schultz, 2008) problems where "a learner tries to figure out what a teacher wants them to do" (Grizou et al., 2013; Cederborg & Oudeyer, 2014). Specifically, the challenge of *Brain Computer Interfaces* (BCI), where users use brain signals to control virtual and robotic agents in sequential tasks (Katyal et al., 2014; deBettencourt et al., 2015; Mishra & Gazzaley, 2015; Muñoz-Moldes & Cleeremans, 2020; Chiang et al., 2021), is well captured by the ABP. In BCIs, (3) is identified as the calibration problem and is usually tackled with supervised learning to learn a mapping between signals and meanings. As this calibration phase is often laborious and impractical for users, current approaches investigate calibration-free solutions where the mapping is learned interactively (Grizou et al., 2014; Xie et al., 2021). Yet, these works consider that the user (i.e. the architect) is fixed, in the sense that it does not adapt to the agent (i.e. the builder) and uses a set of pre-defined instructions (or feedback) meanings that the agent must learn to map to signals. In our ABP formulation however, the architect is dynamic and, as interactions unfold, must learn to best guide a learning builder by tuning the meanings of instructions according to the builder's reactions. In that sense, ABP provides a more complete computational model of agent-agent or human-agent interaction.

With all these constraints in mind, we propose Architect Builder Iterated Guiding (ABIG), an algorithmic solution to ABP when both agents are AIs. ABIG is inspired by the field of experimental semiotics and relies on two high-level interaction priors: *shared intent* and *interaction frames*. Shared intent refers to the fact that, although the builder ignores the objective of the task to fulfill, it will assume that its objective is aligned with the architect's. This assumption is characteristic of cooperative tasks and shown to be a necessary condition for the emergence of communication both in practice (Foerster et al., 2016; Cao et al., 2018) and in theory (Crawford & Sobel, 1982). Specifically, the builder should assume that the architect is guiding it towards a shared objective. Knowing this, the builder must reinforce the behavior it displays when guided by the architect. We show that the builder can efficiently implement this by using imitation learning on its own guided behavior. Because the builder imitates itself, we call it self-imitation. The notion of *interaction frames* (also called *pragmatic frames*) states that agents that interact in sequence can more easily interpret the interaction history (Bruner, 1985; Vollmer et al., 2016). In ABIG, we consider two distinct interaction frames. These are stationary which means that when one agent learns, the other agent's behavior is fixed. During the first frame (the modelling frame), the builder is fixed and the architect learns a model of the builder's message-conditioned behavior. During the second frame (the guiding frame), the architect is fixed and the builder learns to be guided via self-imitation learning.

We show that ABIG results in a low-level, high-frequency, guiding communication protocol that not only enables an architect-builder pair to solve the task at hand, but can also be used to solve unseen tasks. **Our contributions are:**

- The Architect-Builder Problem (ABP), an interactive learning setting to study how artificial agents can simultaneously learn to solve a task and derive a communication protocol.
- Architect-Builder Iterated Guiding (ABIG), an algorithmic solution to the ABP.
- An analysis of ABIG's key learning mechanisms.
- An evaluation of ABIG on a construction environment where we show that ABIG agents evolve communication protocols that generalize to unseen harder tasks.
- A detailed analysis of ABIG's learning dynamics and impact on the mutual information between messages and actions (in the Supplementary Material).

## 2 PROBLEM DEFINITION

**The Architect-Builder Problem.** We consider a multi-agent setup composed of two agents: an architect and a builder. Both agents observe the environment state $s$ but only the architect knows the goal at hand. The architect cannot take actions in the environment but receives the environmental reward $r$ whereas the builder does not receive any reward and has thus no knowledge about the task at hand. In this asymmetrical setup, the architect can only interact with the builder through a communication signal $m$ sampled from its policy $\pi_A(m|s)$. These messages, that have no a priori meanings, are received by the builder which acts according to its policy $\pi_B(a|s,m)$. This makes the environment transition to a new state $s'$ sampled from $P_E(s'|s,a)$ and the architect receives reward $r'$. Messages are sent at every time-step. The CoCo game that inspired ABP is sketched in Figure 1(a) while the overall architect-builder-environment interaction diagram is given in Figure 1(b). The differences between the ABP setting and the MARL and IRL settings are illustrated in Figure 8.

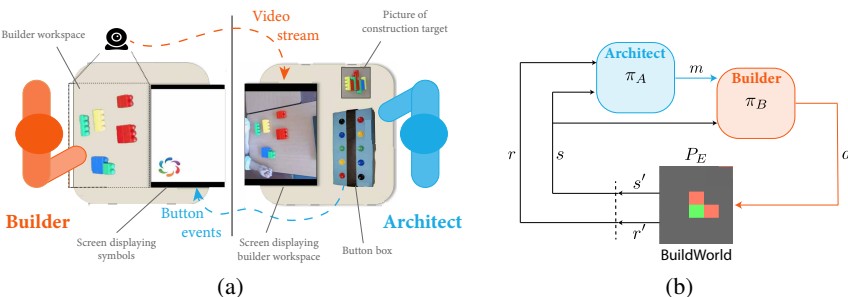

(a)          (b)

**Figure 1:** (a) **Schematic view of the CoCo Game (the inspiration for ABP).** The architect and the builder should collaborate in order to build the construction target while located in different rooms. The architecture has a picture of the target while the builder has access to the blocks. The architect monitors the builder workspace via a camera (video stream) and can communicate with the builder only through the use of 10 symbols (button events). (b) **Interaction diagram between the agents and the environment in our proposed ABP.** The architect communicates messages ($m$) to the builder. Only the builder can act ($a$) in the environment. The builder conditions its action on the message sent by the builder ($\pi_B(a|s,m)$). The builder never perceives any reward from the environment. A schematic view of the equivalent ABP problem is provided in Figure 7(b).

**BuildWorld.** We conduct our experiments in *BuildWorld*. BuildWorld is a 2D construction grid-world of size $(w \times h)$. At the beginning of an episode, the agent and $N_b$ blocks are spawned at different random locations. The agent can navigate in this world and grasp blocks by activating its gripper while on a block. The action space $\mathcal{A}$ is discrete and include a "do nothing" action ($|\mathcal{A}| = 6$). At each time step, the agent observes its position in the grid, its gripper state as well as the position of all the blocks and if they are grasped ($|\mathcal{S}| = 3 + 3N_b$).

**Tasks.** BuildWorld contains 4 different training tasks: 1) 'Grasp': The agent must grasp any of the blocks; 2) 'Place': The agent must place any block at a specified location in the grid; 3/4) 'H-Line/V-line': The agent must place all the blocks in a horizontal/vertical line configuration. BuildWorld also has a harder fifth testing task, '6-blocks-shapes', that consists of more complex configurations and that is used to challenge an algorithm's transfer abilities. For all tasks, rewards are sparse and only given when the task is completed.

This environment encapsulates the interactive learning challenge of ABP while removing the need for complex perception or locomotion. In the RL setting, where the same agent acts and receives

rewards, this environment would not be very impressive. However, it remains to be shown that the tasks can be solved in the setting of ABP (with a reward-less builder and an action-less architect).

**Communication.** The architect guides the builder by sending messages $m$ which are one-hot vectors of size $|\mathcal{V}|$ ranging from 2 to 72, see Suppl. Section B.3 for the impact of this parameter.

**Additional Assumptions.** In order to focus on the architect-builder interactions and the learning of a shared communication protocol, the architect has access to $P_E(s'|s,a)$ and to the reward function $r(s,a)$ of the goal at hand. This assumes that, if the architect were to act in the environment instead of the builder, it would be able to quickly figure out how to solve the task. This assumption is compatible with the CoCo game experiment (Vollmer et al., 2014) where humans participants, and in particular the architects, are known to have such world models.

## 3 ABIG: ARCHITECT-BUILDER ITERATED GUIDING

### 3.1 ANALYTICAL DESCRIPTION

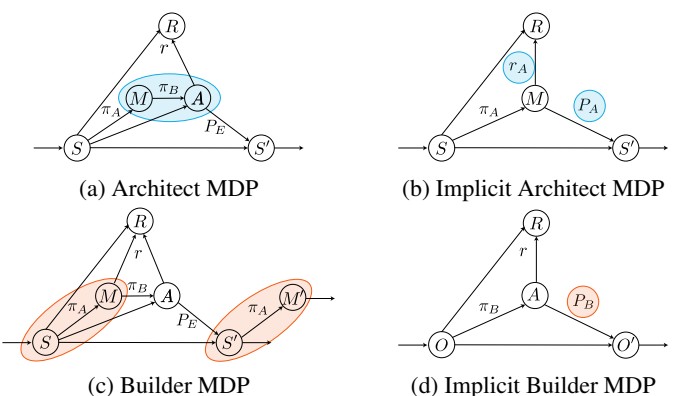

(a) Architect MDP       (b) Implicit Architect MDP

(c) Builder MDP       (d) Implicit Builder MDP

Figure 2: **Agent's Markov Decision Processes.** Highlighted regions refer to MDP coupling. (a) The architect's transitions and rewards are conditioned by the builder's policy $\pi_B$. (b) Architect's MDP where transition and reward models implicitly account for builder's behavior. (c-d) The builder's transition model depends on the architect's message policy $\pi_A$. The builder's learning signal $r$ is unknown.

**Agents-MDPs.** In the Architect-Builder Problem, agents are operating in different, yet coupled, MDPs. Those MDPs depend on their respective point of view (see Figure 2). From the point of view of the architect, messages are actions that influence the next state as well as the reward (see Figure 2 (a)). The architect knows the environment transition function $P_E(s'|s,a)$ and $r(s,a)$, the true reward function associated with the task that does not depend explicitly on messages. It can thus derive the effect of its messages on the builder's actions that drive the reward and the next states (see Figure 2 (b)). On the other hand, the builder's state is composed of the environment state and the message, which makes estimating state transitions challenging as one must also capture the message dynamics (see Figure 2 (c)). Yet, the builder can leverage its knowledge of the architect picking messages based on the current environment state. The equivalent transition and reward models, when available, are given below (see derivations in Suppl. Section A).

$$
\left.\begin{aligned}
P_A(s'|s,m) &= \sum_{a\in\mathcal{A}} \tilde{\pi}_B(a|s,m) P_E(s'|a,s) \\
r_A(s,m) &= \sum_{a\in\mathcal{A}} \tilde{\pi}_B(a|s,m) r(s,a)
\end{aligned}\right\} \quad \text{with} \quad \tilde{\pi}_B(a|s,m) \triangleq P(a|s,m) \tag{1}
$$

$$
P_B(s',m'|s,m,a) = \tilde{\pi}_A(m'|s') P_E(s'|s,a) \quad \text{with} \quad \tilde{\pi}_A(m'|s') \triangleq P(m'|s') \tag{2}
$$

where subscripts $A$ and $B$ refer to the architect and the builder, respectively. $\tilde{x}$ denotes that $x$ is unknown and must be approximated. From the builder's point of view, the reward – denoted $\tilde{r}$ – is unknown. This prevents the use of classical RL algorithms.

**Shared Intent and Interaction Frames.** It follows from Eq. (1) that, provided that it can approximate the builder's behavior, the architect can compute the reward and transition models of its MDP. It can then use these to derive an optimal message policy $\pi_A^*$ that would maximize its objective:

$$\pi_A^* = \underset{\pi_A}{\operatorname{argmax}}\, G_A = \underset{\pi_A}{\operatorname{argmax}}\, \mathbb{E}[\sum_t \gamma^t r_{A,t}] \tag{3}$$

$\gamma \in [0,1]$ is a discount factor and the expectation can be thought of in terms of $\pi_A$, $P_A$ and the initial state distribution. However, the expectation can also be though in terms of the corresponding trajectories $\tau \triangleq \{(s, m, a, r)_t\}$ generated by the architect-builder interactions. In other words, when using $\pi_A^*$ to guide the builder, the architect-builder pair generates trajectories that maximizes $G_A$. The builder has no reward signal to maximize, yet, it relies on a shared intent prior and assumes that its objective is the same as the architect's one:

$$G_B = G_A = \mathbb{E}_\tau[\sum_t \gamma^t r_{A,t}] = \mathbb{E}_\tau[\sum_t \gamma^t \tilde{r}_t] \tag{4}$$

where the expectations are taken with respect to trajectories $\tau$ of architect-builder interactions. Therefore, under the shared intent prior, architect-builder interactions where the architect uses $\pi_A^*$ to maximize $G_A$ also maximize $G_B$. This means that the builder can interpret these interaction trajectories as demonstrations that maximize its unknown reward function $\tilde{r}$. Consequently, the builder can reinforce the desired behavior – towards which the architect guides it – by performing self-Imitation Learning[1] on the interaction trajectories $\tau$.

Note that in Eq. (1), the architect's models can be interpreted as expectations with respect to the builder's behavior. Similarly, the builder's objective depends on the architect's guiding behavior. This makes one agent's MDP highly non-stationary and the agent must adapts its behavior if the other agent's policy changes. To palliate to this, agents rely on interaction frames which means that, when one agent learns, the other agent's policy is fixed to restore stationarity. The equivalent MDPs for the architect and the builder are respectively $\mathcal{M}_A = \langle \mathcal{S}, \mathcal{V}, P_A, r_A, \gamma \rangle$ and $\mathcal{M}_B = \langle \mathcal{S} \times \mathcal{V}, \mathcal{A}, P_B, \emptyset, \gamma \rangle$. Finally, $\pi_A : \mathcal{S} \mapsto \mathcal{V}$, $P_A : \mathcal{S} \times \mathcal{V} \mapsto [0, 1]$, $r_A : \mathcal{S} \times \mathcal{V} \mapsto [0, 1]$, $\pi_B : \mathcal{S} \times \mathcal{V} \mapsto \mathcal{A}$ and $P_B : \mathcal{S} \times \mathcal{V} \times \mathcal{A} \mapsto [0, 1]$ where $\mathcal{S}, \mathcal{A}$ and $\mathcal{V}$ are respectively the sets of states, actions and messages.

## 3.2 PRACTICAL ALGORITHM

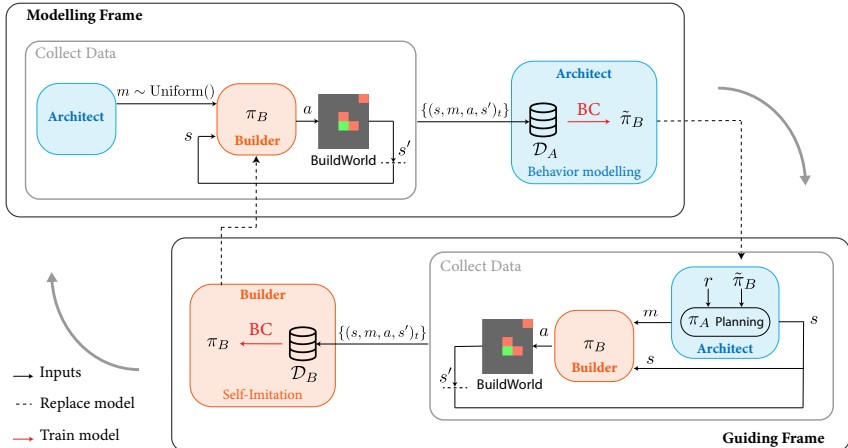

**Figure 3: Architect-Builder Iterated Guiding.** Agents iteratively interact through the modelling and guiding frames. In each frame, one agent collects data and improves its policy while the other agent's behavior is fixed.

ABIG iteratively structures the interactions between a builder-architect pair into interaction frames. Each iteration starts with a *modelling frame* during which the architect learns a model of the builder. Directly after, during the *guiding frame*, the architect leverages this model to produce messages that

---

[1] not to be confused with Oh et al. (2018) which is an off-policy actor-critic algorithm promoting exploration in single-agent RL.

guide the builder. On its side, the builder stores the guiding interactions to train and refine its policy $\pi_B$. The interaction frames are described below. The algorithm is illustrated in Figure 3 and the pseudo-code is reported in **Algorithm 1 in Suppl. Section A.3**.

**Modelling Frame.** The architect records a data-set of interactions $\mathcal{D}_A \triangleq \{(s, m, a, s')_t\}$ by sending random messages $m$ to the builder and observing its reaction. After collecting enough interactions, the architect learns a model of the builder $\tilde{\pi}_B$ using *Behavioral Cloning* (BC) (Pomerleau, 1991).

**Guiding Frame.** During the guiding frame, the architect observes the environment states $s$ and produces messages so as to maximize its return (see Eq. 3). The policy of the architect is a Monte Carlo Tree Search Algorithm (MCTS) (Kocsis & Szepesvári, 2006) that searches for the best message by simulating the reaction of the builder using $\tilde{a} \sim \tilde{\pi}_B(\cdot|m, s)$ alongside the dynamics and reward models. During this frame, the builder stores the interactions in a buffer $\mathcal{D}_B \triangleq \{(s, m, a, s')_t\}$. At the end of the guiding frame, the builder self-imitates by updating its policy $\pi_B$ with BC on $\mathcal{D}_B$.

**Practical Considerations.** All models are parametrized by two-hidden layer 126-units feedforward ReLu networks. BC minimizes the cross-entropy loss with Adam optimizer (Kingma & Ba, 2015). Networks are re-initialized before each BC training. The architect's MCTS uses Upper-Confidence bound for Trees and relies on heuristics rather than Monte-Carlo rollouts to estimate the value of states. For more details about training, MCTS and hyper-parameters please see Suppl. Section A.3.

The resulting method (ABIG) is general and can handle a variety of tasks while not restricting the kind of communication protocol that can emerge. Indeed, it only relies on a few high-level priors, namely, the architect's access to environment models, shared intent and interaction frames.

In addition to ABIG we also investigate two control settings: ABIG *-no-intent* – the builder interacts with an architect that disregards the goal and therefore sends random messages during training. At evaluation, the architect has access to the exact model of the builder ($\tilde{\pi}_B = \pi_B$) and leverages it to guide it towards the evaluation goal (the architect no longer disregards the goal). And *random* – the builder takes random actions. The comparison between ABIG and ABIG-no-intent measures the impact of doing self-imitation on guiding versus on non-guiding trajectories. The random baseline is used to provide a performance lower bound that indicates the task's difficulty.

### 3.3 UNDERSTANDING THE LEARNING DYNAMICS

Architect-Builder Iterated Guiding relies on two steps. First, the architect selects *favorable* messages, i.e. messages that maximize the likelihood of the builder picking optimal actions with respect to the architect's reward. Then, the builder does self-imitation and reinforces the guided behavior by maximizing the likelihood of the corresponding messages-actions sequence under its policy. The message-to-action associations (or preferences) are encoded in the builder's policy $\pi_B(a|s, m)$. Maximum likelihood assumes that actions are initially equiprobable for a given message. Therefore, actions under a message that is not present in the data-set ($\mathcal{D}_B$) remains so. In other words, if the builder never observes a message, it assumes that this message is equally associated with all the possible actions. This enables the builder to *forget* past message-to-action associations that are not used – and thus not reinforced – by the architect. In practice, initial uniform likelihood is ensured by resetting the builder's policy network before each self-imitation. The architect can leverage the forget mechanism to erase unfavorable associations until a favorable one emerges. Such favorable associations can then be reinforced by the architect-builder pair until it is made deterministic. The *reinforcement* process of favorable associations is also enabled by the self-imitation phase. Indeed, for a given message $m$, the self-imitation objective for $\pi$ on a data-set $\mathcal{D}$ collected using $\pi$ is:

$$J(m, \pi) = -\sum_{a \sim \mathcal{D}} \log \pi(a|m) \approx \mathbb{E}_{a \sim \pi(\cdot|m)}[-\log \pi(a|m)] \approx H[\pi(\cdot|m)] \tag{5}$$

where $H$ stands for the entropy of a distribution. Therefore, maximizing the likelihood in this case results in minimizing the entropy of $\pi(\cdot|m)$ and thus reinforces the associations between messages and actions. Using these mechanisms the architect can adjust the policy of the builder until it becomes *controllable*, i.e. deterministic (strong preferences over actions for a given message) and flexible (varied preferences across messages). Conversely, in the case of ABIG-no-intent, the architect does not guide the builder and simply sends messages at random. Favorable and unfavorable messages are thus sampled alike which prevents the forget mechanism to undo unfavorable message-to-action associations. Consequently in that case, self-imitation tends to simply reinforce initial

builder's preferences over actions making the controllability of the builder policy depend heavily on the initial preferences. We illustrate the above learning mechanisms in Suppl. Section A.4 by applying ABIG to a simple instantiation of the ABP. Figure 9 and Figure 11 confirm that ABIG uses the forget and reinforcement mechanisms to circumvent the unfavorable initial conditions while ABIG-no-intent simply reinforces them. Eventually, Figure 11 reports that ABIG always reaches 100% success rate regardless of the initial conditions while ABIG-no-intent success rate depends on the initial preferences (only 3% when they are unfavorable).

Interestingly, the emergent learning mechanisms discussed here are reminiscent of the amplification and self-enforcement of random fluctuations in naming games (Steels, 1995). In naming games however, the self-organisation of vocabularies are driven by each agent maximizing its communicative success whereas in our case the builder has no external learning signal and simply self-imitates.

## 4 RELATED WORK

This work is inspired by experimental semiotics (Galantucci & Garrod, 2011) and in particular Vollmer et al. (2014) that studied the CoCo game with human subjects as a key step towards understanding the underlying mechanisms of the emergence of communication. Here we take a complementary approach by defining and investigating solutions to the ABP, a general formulation of the CoCo game where both agents are AIs.

Recent MARL work (Lowe et al., 2017; Woodward et al., 2020; Roy et al., 2020; Ndousse et al., 2021), investigate how RL agents trained in the presence of other agents leverage the behaviors they observe to improve learning. In these settings, the other agents are used to build useful representation or gain information but the main learning signal of every agent remains a ground truth reward.

Feudal Learning (Dayan & Hinton, 1992; Kulkarni et al., 2016; Vezhnevets et al., 2017; Nachum et al., 2018; Ahilan & Dayan, 2019) investigate a setting where a manager sets the rewards of workers to maximize its own return. In this Hierarchical setting, the manager interacts by directly tweaking the workers' learning signal. This would be unfeasible for physically distinct agents, hence those methods are restricted to single-agent learning. On the other hand, ABP considers separate agents, that must hence communicate by influencing each other's observations instead of rewards signals.

Inverse Reinforcement Learning (IRL) (Ng et al., 2000) and Imitation Learning (IL) (Pomerleau, 1991) have been investigated for HRI when it is challenging to specify a reward function. Instead of defining rewards, IRL and IL rely on expert demonstrations. Hadfield-Menell et al. (2016) argue that learning from expert demonstrations is not always optimal and investigate how to produce instructive demonstrations to best teach an apprentice. Crucially, the expert is aware of the mechanisms by which the apprentice learns, namely RL on top of IRL. This allows the expert to assess how its demonstrations influence the apprentice policy, effectively reducing the problem to a single agent POMDP. In our case however, the architect and the builder do not share the same action space which prevents the architect from producing demonstrations. In addition, the architect ignores the builder's learning process which makes the simplification to a single agent teacher problem impossible.

In essence, the ABP is closest to works tackling the calibration-free BCI control problem (Grizou et al., 2014; Xie et al., 2021). Yet, these works both consider that the architect sends messages after the builder's actions and thus enforce that the feedback conveys a reward. Crucially, the architect does not learn and communicates with a fixed mapping between feedback and pre-defined meanings ("correct" vs. "wrong"). Those meanings are known to the builder and it simply has to learn the mapping between feedback and meaning. In our case however, the architect communicates before the builder's action and thus rather gives instructions than feedback. Additionally, the builder has no a priori knowledge of the set of possible meanings and the architect adapts those to the builder's reaction. Finally, Grizou et al. (2013) handles both feedback and instruction communications but relies on known task distribution and set of possible meanings. In terms of motivations, previous works are interested in one robot figuring out a fixed communication protocol while we train two agents to collectively emerge one.

Our BuildWorld resembles GridLU proposed by Bahdanau et al. (2019) to analyze reward modelling in language-conditioned learning. However, their setting is fundamentally different to ours as it investigates single agent goal-conditioned IL where goals are predefined episodic linguistic instructions labelling expert demonstrations. Nguyen et al. (2021) alleviate the need for expert demonstra-

tions by introducing an interactive teacher that provides descriptions of the learning agent's trajectories. In this HRI setting, the teacher still follows a fixed pre-defined communication protocol known by the learner: messages are activity descriptions. Our ABP formulation relates to the Minecraft Collaborative Building Task (Narayan-Chen et al., 2019) and the IGLU competition (Kiseleva et al., 2021); however, they do not consider emergent communication. Rather, they focus on generating architect utterances by leveraging a human-human dialogues corpus to learn pre-established meanings expressed in natural language. Conversely, in ABP both agents learn and must evolve the meanings of messages while solving the task without relying on any form of demonstration.

## 5 RESULTS

In the following sections, success rates (sometimes referred as scores) are averaged over 10 random seeds and error bars are $\pm$2SEM with SEM the Standard Error of the Mean. If not stated otherwise, the grid size is $(5 \times 6)$, contains three blocks ($N_b = 3$) and the vocabulary size is $|\mathcal{V}| = 18$.

**ABIG's learning performances.** We apply ABIG to the four learning tasks of BuildWorld and compare it with the two control settings: ABIG-no-intent (no guiding during training) and random (builder takes random actions). Figure 4 reports the mean success rate on the four tasks defined in Section 2. First, we observe that ABIG significantly outperforms the control conditions on all tasks. Second, we notice that on the simpler 'grasp' task ABIG-no-intent achieves a satisfactory mean score of 0.77$\pm$0.03. This is consistent with the learning dynamic analysis provided in Sup. Section A.4 that shows that, in favorable settings, a self-imitating builder can develop a reasonably controllable policy (defined in Section 3.3) even if it learns on non-guiding trajectories. Nevertheless, when the tasks get more complicated and involve placing objects or drawing lines, the performances of ABIG-no-intent drop significantly whereas ABIG continues to achieve high success rates ($> 0.8$). This demonstrates that ABIG enables a builder-architect pair to successfully agree on a communication protocol that makes the builder's policy controllable and enables the architect to efficiently guide it.

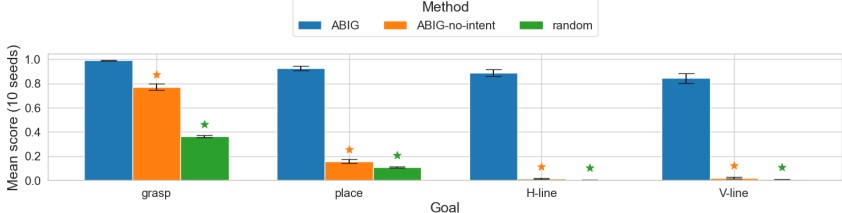

**Figure 4:** Methods performances (stars indicate significance with respect to ABIG model according to Welch's $t$-test with null hypothesis $\mu_1 = \mu_2$, at level $\alpha = 0.05$). ABIG outperforms control baselines on all goals.

**ABIG's transfer performances.** Building upon previous results, we propose to study whether a learned communication protocol can transfer to new tasks. The architect-builder pairs are trained on a single task and then evaluated without retraining on the four tasks. In addition, we include 'all-goals': a control setting in which the builder learns a single policy by being guided on all four goals during training. Figure 5 shows that, on all training tasks except 'grasp', ABIG enables a transfer performance above 0.65 on all testing tasks. Notably, training on 'place' results in a robust communication protocol that can be used to solve the other tasks with a success rate above 0.85, being effectively equivalent as training on 'all-goals' directly. This might be explained by the fact that placing blocks at specified locations is an atomic operation required to build lines.

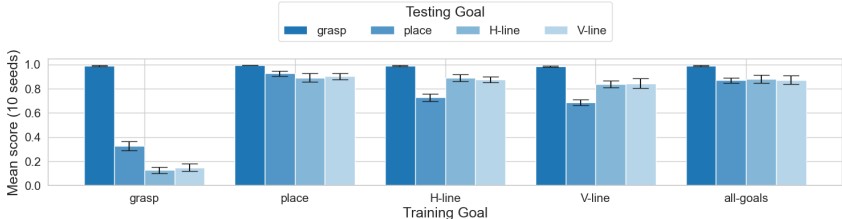

**Figure 5:** ABIG transfer performances without retraining depending on the training goal. ABIG agents learn a communication protocol that transfers to new tasks. Highest performances reached when training on 'place'.

**Challenging ABIG's transfer abilities.** Motivated by ABIG's transfer performances, we propose to train it on the 'place' task in a bigger grid ($6 \times 6$) with $N_b = 6$ and $|\mathcal{V}| = 72$. Then, without retraining, we evaluate it on the '6-block-shapes' task[2] that consists in constructing the shapes given in Figure 6. The training performance on 'place' is $0.96 \pm 0.02$ and the transfer performance on the '6-block-shapes' is $0.85 \pm 0.03$. This further demonstrates ABIG's ability to derive robust communication protocols that can solve more challenging unseen tasks.

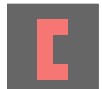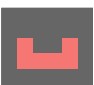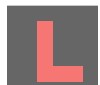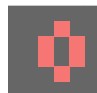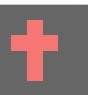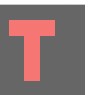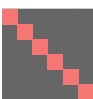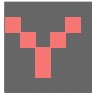

**Figure 6:** 6-block-shapes that ABIG can construct in transfer mode when trained on the 'place' task.

**Additional experiments.** The Supplementary Materials contains the following experiments:

- Figures 9, 10 and 11 analyse the builder's message-to-action preferences. They illustrate ABIG's learning mechanisms (forget and reinforce) and compare them to ABIG-no-intent's.
- Figure 12 shows that, as the communication protocol settles, the message-action mutual information becomes greater than the state-action mutual information which is a desirable feature for the emergence of communication.
- Figure 13 reports ABIG outperforming complementary baselines.
- Figure 14 shows ABIG's performance increasing with the vocabulary size, suggesting that with more messages available, the architect can more efficiently refer to the desired action.

## 6 DISCUSSION AND FUTURE WORK

This work formalizes the ABP as an interactive setting where learning must occur without explicit reinforcement, demonstrations or a shared language. To tackle ABP, we propose ABIG: an algorithm allowing to learn how to guide and to be guided. ABIG is based only on two high-level priors to communication emergence (shared intent and interactions frames). ABP's general formulation allows us to formally enforce those priors during learning. We study their influence through ablation studies, highlighting the importance of shared intent achieved by doing self-imitation on guiding trajectories. When performed in interaction frames, this mechanism enables agents to evolve a communication protocol that allows them to solve all the tasks defined in BuildWorld. More impressively, we find that communication protocols derived on a simple task can be used to solve harder, never-seen goals.

Our approach has several limitations which open up different opportunities for further work. First, ABIG trains agents in a stationary configuration which implies doing several interaction frames. Each interaction frame involves collecting numerous transitions. Thus, ABIG is not data efficient. A challenging avenue would be to relax this stationarity constraint and have agents learn from buffers containing non-stationary data with obsolete agent behaviors. Second, the builder remains dependent on the architect's messages even at convergence. Using a Vygotskian approach (Colas et al., 2020; 2021), the builder could internalize the guidance from the architect to become autonomous in the task. This could, for instance, be achieved by having the builder learn a model of the architect's message policy once the communication protocol has converged.

Because we present the first step towards interactive agents that learn in the ABP, our method uses simple tools (feed-forward networks and self-imitation learning). It is however important to note that our proposed formulation of the ABP can support many different research directions. Experimenting with agents' models could allow for the investigation of other forms of communication. One could, for instance, include memory mechanisms in the models of agents in order to facilitate the emergence of retrospective feedback, a form of emergent communication observed in Vollmer et al. (2014). ABP is also compatible with low-frequency feedback. As a further experiment in this direction, one could penalize the architect for sending messages and assess whether a pair can converge to higher-level meanings. Messages could also be composed of several tokens in order to allow for the emergence of compositionality. Finally, our proposed framework can serve as a testbed to study the fundamental mechanisms of emergent communication by investigating the impact of high level communication priors from experimental semiotics.

---

[2] For rollouts see `https://sites.google.com/view/architect-builder-problem/`

# 7 ETHICS STATEMENT

This work investigates a novel interactive approach to autonomous agent learning and proposes a fundamental learning setting. This work does not directly present sensitive applications or data. While the implications of autonomous agents learning are not trivial, we do not aim here at discussing the general impact of autonomous agents. Rather, we focus on discussing how the proposed learning setting contrasts with more classical supervisions such as reward signals and demonstrations. By proposing an iterative and interactive learning setting, the Architect-Builder Problem (ABP) promotes a finer control over learned behavior than designing rewards or demonstrations. Indeed, the behavior is constantly evaluated and refined throughout learning as interactions unfold. Still, in this process it is essential to keep in mind the importance of the architect as it is the agent that judges if the learned behavior is satisfactory.

# 8 REPRODUCIBILITY STATEMENT

We ensure the reproducibility of the experiments presented in this work by providing our code[3]. Additional information regarding the methods and hyper-parameters can be found in Section 3.2 and in the Suppl. Section A.3. We ensure the statistical significance of our experimental results by using 10 random seeds, reporting the standard error of the mean and using Welch's $t$-test. Finally, we propose complete analytical derivations in Suppl. Section A.

## ACKNOWLEDGMENT

The authors thank Erwan Lecarpentier for valuable advice on Monte-Carlo Tree Search as well as Compute Canada and GENCI-IDRIS (Grant 2020-A0091011996) for providing computing resources. Derek and Paul acknowledge support from the NSERC Industrial Research Chair program. Tristan Karch is partly funded by the French Ministère des Armées - Direction Générale de l'Armement.

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

# SUPPLEMENTARY MATERIAL

This Supplementary Material provides additional derivations, implementation details and results. More specifically:

- Section A proposes derivations, implementation details and analysis related to our method.
  - Subsection A.1 provides additional diagrams illustrating the ABP problem and its position with respect to related settings.
  - Subsection A.2 proposes the full derivation of the agents' MDP.
  - Subsection A.3 proposes our methods pseudo-code, algorithmic implementation details (for BC and MCTS), hyper-parameters and compute resources.
  - Subsection A.4 proposes analysis that explore our method's learning mechanisms.
  - Subsection A.5 discusses the differences between ABP and Hierarchical/Feudal Reinforcement Learning.
- Section B provides additional results.
  - Subsection B.1 monitors the builder's behavior properties as training progresses.
  - Subsection B.2 compares our method to additional baselines.
  - Subsection B.3 analyses the impact of the vocabulary size on the learning performance.

## A   SUPPLEMENTARY METHODS

### A.1   SUPPLEMENTARY SKETCHES

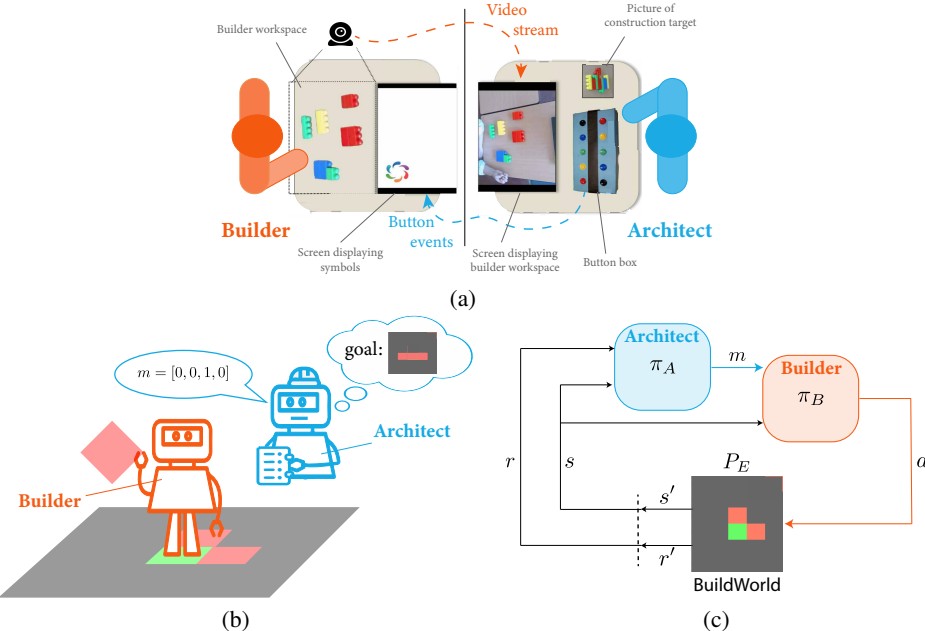

(a)

(b)                                         (c)

**Figure 7:** (a) **Schematic view of the CoCo Game.** The architect and the builder should collaborate in order to build the construction target while located in different rooms. The architecture has a picture of the target while the builder has access to the blocks. The architect monitors the builder workspace via a camera (video stream) and can communicate with the builder only through the use of 10 symbols (button events). (b) **Schematic view of the Architect-Builder Problem.** The architect must learn how to use messages to guide the builder while the builder needs to learn to make sense of the messages in order to be guided by the architect. (c) **Interaction diagram between the agents and the environment in our proposed ABP.** The architect communicates messages ($m$) to the builder. Only the builder can act ($a$) in the environment. The builder conditions its action on the message sent by the builder ($\pi_B(a|s,m)$). The builder never perceives any reward from the environment

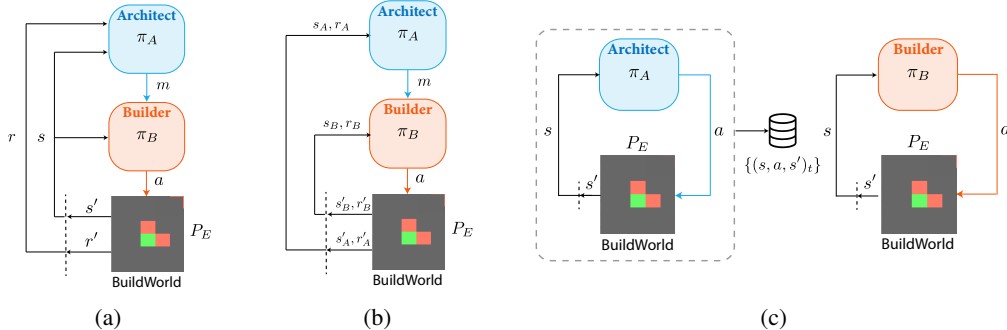

**Figure 8:** (a) **Vertical view of the interaction diagram between the agents and the environment in our proposed ABP.** Only the architect perceives a reward signal $r$; (b) **Interaction diagram for a standard MARL modelization.** Both the architect and the builder have access to environmental rewards $r_A$ and $r_B$. Which would contradict the fact that the builder ignores everything about the task at hand; (c) **Inverse Reinforcement Learning modelization of the ABP.** The architect needs to provide demonstrations. The architect does not exchange messages with the builder. The builder relies on the demonstrations $\{(s, a, s')_t\}$ to learn the desired behavior.

## A.2 ANALYTICAL DESCRIPTION

**Transition Probabilites from the architect point of view** Using the laws of total probabilities and conditional probabilities we have:

$$
\begin{aligned}
P_A(s'|s, m) &= \sum_{a \in \mathcal{A}} P(s', a|s, m) \\
&= \sum_{a \in \mathcal{A}} P(s'|a, s, m) P(a|s, m) \\
&= \sum_{a \in \mathcal{A}} P_E(s'|a, s) \tilde{\pi}_b(a|s, m)
\end{aligned}
\tag{6}
$$

Where the final equality uses the knowledge that next-states only depends on states and builder's actions.

**Reward function from the architect point of view**

$$
\begin{aligned}
r_A(s, m, s') &\triangleq \mathbb{E}[R|s, m, s'] \\
&= \int_{\mathbb{R}} r P(r|s, m, s') dr \\
&= \int_{\mathbb{R}} r \sum_{a \in \mathcal{A}} P(r, a|s, m, s') dr \\
&= \int_{\mathbb{R}} r \sum_{a \in \mathcal{A}} P(r|s, m, a, s') P(a|s, m, s') dr \\
&= \int_{\mathbb{R}} r \sum_{a \in \mathcal{A}} P(r|s, a, s') \tilde{\pi}_b(a|s, m) dr \\
&= \sum_{a \in \mathcal{A}} \tilde{\pi}_b(a|s, m) \int_{\mathbb{R}} r P(r|s, a, s') dr \\
&= \sum_{a \in \mathcal{A}} \tilde{\pi}_b(a|s, m) r(s, a, s')
\end{aligned}
\tag{7}
$$

**Transition function from the builder point of view**

$$
\begin{aligned}
P(s', m'|s, m, a) &= P(m'|s', s, m, a)P(s'|s, m, a) \\
&= P(m'|s')P(s'|s, a) \\
&= \tilde{\pi}_A(m'|s')P_E(s'|s, a)
\end{aligned}
\tag{8}
$$

## A.3 PRACTICAL ALGORITHM

---
**Algorithm 1:** Architect-Builder Iterated Guiding (ABIG)

---
**Require:** randomly initialized builder policy $\pi_B$, reward function $r$, transition function $P_E$, BC
    algorithm, MCTS algorithm
    **for** $i$ in range($N_{iterations}$) **do**
        MODELLING FRAME:
            **for** $e$ in range($N_{collect}/2$) **do**
                Architect populates $\mathcal{D}_A$ using $m \sim$ Uniform() and observing $a \sim \pi_B(\cdot|s, m)$
            **end for**
            Architect learns $\tilde{\pi}_B(a|s, m)$ on $\mathcal{D}_A$ with BC
            Architect sets $\pi_A(m|s) \triangleq$ MCTS($r, \tilde{\pi}_B, P_E$)
            Architect flushes $\mathcal{D}_A$
        GUIDING FRAME:
            **for** $e$ in range($N_{collect}/2$) **do**
                Builder populates $\mathcal{D}_B$ using $\pi_B$ while guided by Architect, i.e. $m \sim \pi_A(\cdot|s)$
            **end for**
            Builder learns $\pi_B(a|s, m)$ on $\mathcal{D}_B$ with BC
            Builder flushes $\mathcal{D}_B$
    **end for**
    Architect runs one last Modelling Frame
**Result:** $\pi_A, \pi_B$

---

**Behavioral Cloning** The data-set is split into training (70%) and validation (30%) sets. If the validation accuracy does not improve during a *wait for* number of epochs the training is early stopped. For a training data-set $\mathcal{D} = \{(s, m, a)\}$ of size $N$ the BC loss to minimize for a policy $\pi_\theta$ parametrized by $\theta$ is given by:

$$
J(\theta) = \frac{1}{N} \sum_{\mathcal{D}} -\log \pi_\theta(a|s, m)
\tag{9}
$$

**Monte-Carlo Tree Search** In the architect's MCTS, nodes are labeled by environment's states and they are expanded by selecting messages. Selecting message $m$ from a node with label $s$ yields a builder action according to the architect's builder model $a \sim \tilde{\pi}_B(a|s, m)$, this sampled action in turn yields the label of the child node according to the environment's transition model $s' \sim P_E(s'|s, a)$. We repeat this process until we select a message that was never selected from the current node or we sample a next state that does not correspond to a child node yet. In both of these cases a new node has to be created. We estimate the value of the new node using an engineered heuristic that estimates the return of an optimal policy $\pi^*(a|s)$ from state $s$. This value is scaled down by a factor 2 to avoid overestimation: the builder's policy may not allow the architect to have it follow $\pi^*$. This estimated value for a newly created node at depth $l$ is back-propagated as a return to parents node at depth $k$ according to:

$$
G^k = \sum_{\tau=0}^{l-1-k} \gamma^\tau r_{k+1+\tau} + \gamma^{l-k}v^l \qquad k = l, ..., 0
\tag{10}
$$

where $r_j$ is the reward collected from node at depth $j$ to child node at depth $j+1$. From a node with label $s$ we select messages according to the Upper Confidence Bound rule:

$$m = \underset{m}{\operatorname{argmax}} \, Q(s,m) + c\sqrt{\frac{\ln \sum_b N(s,b)}{N(s,m)}}$$

$$Q(s,m) = \frac{\sum_i G_i(s,m)}{N(s,m)} \tag{11}$$

where $N(s,m)$ is the number of times message $m$ was selected from the node, $G_i(s,m)$ are the returns obtained from the node when selecting $m$ and $c$ is a constant set to $\sqrt{2}$. When the architect must choose a message from the environment state $s$, its policy $\pi_A(m|s)$ runs the above procedure from a root node labeled with the current environment state $s$. After expanding a budget $b$ of nodes the architect picks the best message to send according to Eq. (11) applied to the root node. It is then possible to reuse the tree for the next action selection or to discard it, if a tree is reused its maximal depth should be constrained.

**Hyper-parameters**

| sampling temperature | samples per iteration | learning rate | number of epochs | batch size |
|---|---|---|---|---|
| 0.5 | 100 | 0.1 | 1000 | 50 |

**Table 1:** Toy experiment hyper-parameters

| budget | reuse tree | max tree depth |
|---|---|---|
| 100 | true | 500 |

**Table 2:** MCTS parameters

| episode len | grid size | reward | message |
|---|---|---|---|
| 40 | 5×6 / (6 × 6) | sparse | one-hot |

| discount factor | episodes per iteration | vocab size | evaluation episode len |
|---|---|---|---|
| 0.95 | 600 | 18 / (72) | 40 / (60) |

**Table 3:** BuildWorld parameters for 3 blocks / (for 6 blocks if different)

| learning rate | number of epochs | batch-size | wait for |
|---|---|---|---|
| $5 \times 10^{-4}$ | 1000 | 256 | 300 |

**Table 4:** Architect's BC parameters on BuildWorld for 3 blocks / (for 6 blocks if different)

| learning rate | number of epochs | batch-size | wait for |
|---|---|---|---|
| $1 \times 10^{-4}$ | 1000 | 256 | 300 |

**Table 5:** Builder's BC parameters on BuildWorld for 3 blocks / (for 6 blocks if different)

Sparse reward means that the architect receives 1 if the goal is achieved and 0 otherwise. Episodes per iterations are equally divided into the modelling and guiding frames. Only the learning rates on BuildWorld were searched over with grid-searches. For BuildWorld with 3 blocks the searched range is $[5 \times 10^{-4}, 1 \times 10^{-4}, 1 \times 10^{-5}]$ for both architect and builder (vocabulary size was fixed at 6). For 'grasp' with 6 blocks the searched range is $[1 \times 10^{-3}, 5 \times 10^{-4}, 1 \times 10^{-4}]$ for the architect and $[5 \times 10^{-4}, 1 \times 10^{-4}, 5 \times 10^{-5}]$ for the builder (vocabulary size was fixed at 72). The other hyper-parameters do not seem to have a major impact on the performance provided that:

- the MCTS hyper-parameters enable an agent that has access to the reward to solve the task.
- there is enough BC epochs to approach convergence.

Regarding the vocabulary size, the bigger the better (see experiments in Figure 14).

**Computing resources** A complete ABIG training can take up to 48 hours on a single modern CPU (`Intel E5-2683 v4 Broadwell @ 2.1GHz`). The presented results require approximately 700 CPU hours. For each training, the main computation cost comes from the MCTS planning during the guiding frames. The self-imitation and behavior modelling steps only account for a small fraction of the computation.

## A.4 INTUITIVE EXPLANATION OF THE LEARNING DYNAMICS

To illustrate the learning mechanisms of ABIG we propose to look at the simplest instantiation of the Architect-Builder Problem: there is one state (thus it can be ignored), two messages $m_1$ and $m_2$ and two possible actions $a_1$ and $a_2$. If the builder chooses $a_1$ it is a loss ($r(a_1) = -1$) but choosing $a_2$ results in a win ($r(a_2) = 1$). Figure 9 displays several iterations of ABIG on this problem when the initial builder's policy is unfavorable ($a_1$ is more likely than $a_2$ for all the messages). During each iteration the architect selects messages in order to maximize the likelihood of the builder picking action $a_2$ and then the builder does self-Imitation Learning by maximizing the likelihood of the corresponding messages-actions sequence under its policy. Figure 9 shows that this process leads to forgetting unfavorable associations until a favorable association emerges and can be reinforced. On the other hand, for ABIG-no-intent in Figure 10, favorable and unfavorable messages are sampled alike which prevents the forget mechanism to undo unfavorable message-to-action associations. Consequently, initial preferences are reinforced.

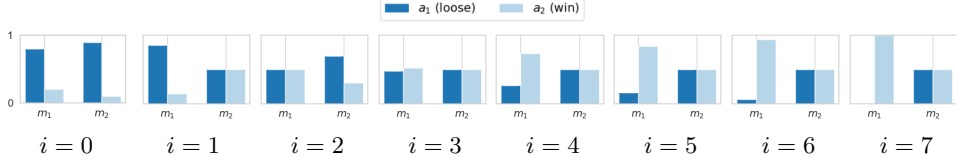

**Figure 9:** ABIG-driven evolution of message-conditioned action probabilities (builder's policy) for a simple problem where the builder must learn to produce action $a_2$. Even under unfavorable initial condition the architect-builder pair eventually manages to associate a message (here $m_1$) with the winning action ($a_2$). Initial conditions are unfavorable since $a_1$ is more likely than $a_2$ for both messages. ($i = 0$) Given the initial conditions, the architect only sends message $m_1$ since it is the most likely to result in action $a_2$. ($i = 1$) the builder guiding data only consisted of $m_1$ message therefore it cannot learn a preference over actions for $m_2$ and both actions are equally likely under $m_2$. The architect now only sends message $m_2$ since it is more likely than $m_1$ at triggering $a_2$. ($i = 2$) Unfortunately, the sampling of $m_1$ resulted in the builder doing more $a_1$ than $a_2$ during the guiding frame and the builder thus associates $m_2$ with $a_1$. The architect tries its luck again but now with $m_1$. ($i = 3$) Eventually, the sampling results in more $a_2$ actions being sampled in the guiding data and the builder now associates $m_1$ to $a_2$. ($i = 4$) and ($i = 5$) The architect can now keep on sending $m_1$ messages to reinforce this association.

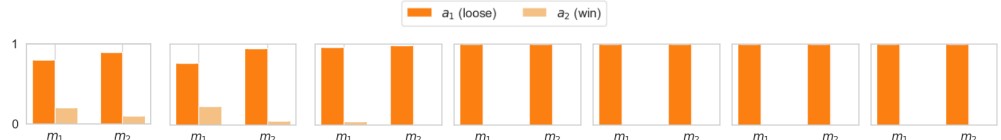

**Figure 10:** ABIG-no-intent driven evolution of message-conditioned action probabilities for a simple problem where builder must learn to produce action $a_2$. Initial conditions are unfavorable since $a_1$ is more likely than $a_2$ for both messages. Without an architect's guiding messages during training, a self-imitating builder reinforces the action preferences of the initial conditions and fails (even when evaluated alongside a knowledgeable architect as both messages can only yield $a_1$).

To further assess how the architect's message choices impact the performance of a self-imitating builder, we compare the distribution of the builder's preferred actions obtained after using ABIG and ABIG-no-intent. We consider three different initial conditions (favorable, unfavorable, intermediate) that are each ran to convergence (meaning that the policy does not change anymore across iterations) for 100 different seeds. Figure 11 displays the resulting distributions of preferred – i.e. most likely – action for each message. When applying ABIG on the toy problem, the pair always reaches a success rate of 100/100 no matter the initial condition. We also observe that – at convergence – the builder never prefers action $a_1$, yet when an action is preferred for a given message, the other message yields no preference over action ($p(a_1|m) = p(a_2|m)$). This is due to the forget mechanism discussed in Section 3.3. The results when applying ABIG-no-intent on the toy problem are much more dependent on the initial condition. In the unfavorable scenario, ABIG-no-intent fails heavily with only 3 seeds succeeding over the 100 experiments. This is due to the fact that, in absence of message guidance from the architect, the builder has high chances to continually reinforce the association between the two messages and $a_1$, therefore losing. However, in rare cases, the builder can inverse the initial message-conditioned probabilities by 'luckily' sampling more often $a_2$ when receiving $m_1$ and win. This only happened 3 times over the 100 seeds. Finally, when initial conditions are more favorable, the self-imitation steps reinforce the association between the messages and $a_2$ which makes the builder prefer $a_2$ for at least one message and enables high success rates (100/100 for favorable and 98/100 for intermediate).

| Unfavorable | Favorable | Intermediate |
|---|---|---|
| | (a) **Initial probabilities** | |
| $P(a_1|m_1) = 0.8, P(a_2|m_1) = 0.2$ | $P(a_1|m_1) = 0.2, P(a_2|m_1) = 0.8$ | $P(a_1|m_1) = 0.9, P(a_2|m_1) = 0.1$ |
| $P(a_1|m_2) = 0.9, P(a_2|m_2) = 0.1$ | $P(a_1|m_2) = 0.1, P(a_2|m_2) = 0.9$ | $P(a_1|m_2) = 0.1, P(a_2|m_2) = 0.9$ |

(b) **ABIG:** Distributions of final preferred action for each message calculated over 100 seeds

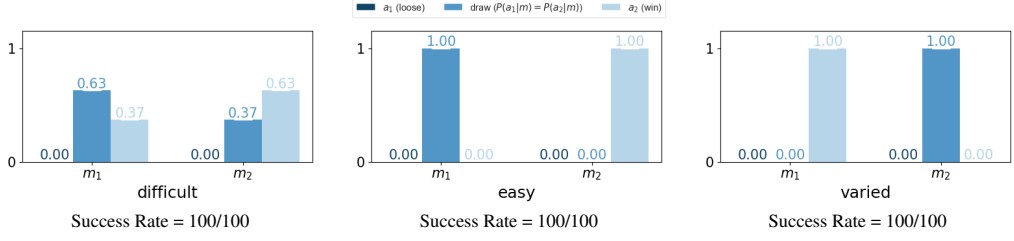

(c) **ABIG-no-intent:** Distributions of final preferred action for each message calculated over 100 seeds

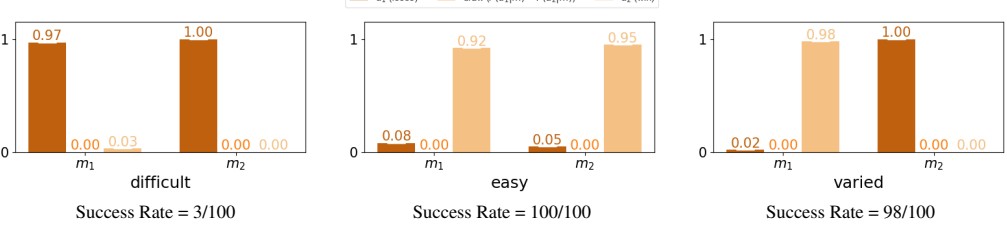

**Figure 11: Toy experiment analysis** (a) Initial conditions: initial probability for each action $a$ given a message $m$; distributions of final builder's preferred actions for each message after applying (b) ABIG and (c) ABIG-no-intent on the toy problem; distributions are calculated over 100 seeds. For each method and each initial condition, we report the success rate obtained by a knowledgeable architect guiding the builder. At evaluation, the architect has access to the builder's model and does not ignore the goal. ABIG always succeeds while ABIG-no-intent's success depends on the initial conditions.

## A.5 RELATED WORK

In this section we develop the differences between ABP and Hierarchical/Feudal Reinforcement Learning more in detail.

Kulkarni et al. (2016) proposes to decompose a RL agent into a two-stage hierarchy with a meta-controller (or manager) setting the goals of a controller (or worker). The meta-controller is trained to select sequences of goals that maximize the environment reward while the controller is trained to maximize goal-conditioned intrinsic rewards. The definition of the goal-space as well as the corresponding hard-coded goal-conditioned reward functions are task-related design choices. In Vezhnevets et al. (2017), the authors propose a more general approach by defining goals as embeddings that directly modulate the worker's policy. Additionally, the authors define intrinsic rewards as the cosine distance between goals and embedded-state deltas (difference between the embedded-state at the moment the goal was given and the current embedded-state). Thus, goals can be interpreted as directions in embedding space. Nachum et al. (2018) build on a this idea but let go of the embedding transformation by considering goals as directions to reach and rewards as distances between state deltas and goals. These works tackle the single-agent learning problem and therefore allow the manager to directly influence the learning signal of the workers. However, in the multi-agent setting where agents are physically distinct, it is not possible for an agent to explicitly tweak another agent's learning algorithm. Instead, agents must communicate by influencing each other's observations instead of intrinsic rewards. Since it is designed to investigate the emergence of communication between agents, ABP lies in this latter multi-agent setting where agents can interact with one-another only through observations. This makes applying Feudal or Hierarchical methods to the ABP unfeasible as they are restricted to worker agents that directly receive rewards. In contrast, in ABP, the reward-less builder observes communication messages that, initially, have arbitrary meaning.

## B  SUPPLEMENTARY RESULTS

### B.1  LEARNING ANALYSIS

The main document shows performances (success rates) of builder-architect pairs at convergence. In this supplementary section we propose to thoroughly study the evolution of the builder's policy in order to provide a deeper analysis of ABIG.

**Metric definition.**   We define three metrics that characterize the builder behavior. We compute these metrics on a constant *Measurement Set* $\mathcal{M}$ made of 6000 randomly sampled states, for each of these states we sample all the possible messages $m \sim \text{Uniform}(\mathcal{V})$ where $\mathcal{V}$ is the set of possible messages. Therefore, $|\mathcal{M}| = 6000 \times |\mathcal{V}|$. The set of possible actions is $\mathcal{A}$ and we denote by $\delta$ the indicator function.

We also define the following distributions:

$$p_S(s) \triangleq \frac{1}{|\mathcal{M}|} \sum_{s' \in \mathcal{M}} \delta(s' == s)$$

$$p_M(m) \triangleq P(m|s) = \frac{1}{|\mathcal{V}|}$$

$$p_{SM}(s,m) \triangleq p_S(s)P(m|s) = p_S(s)p_M(m)$$

$$p_{SMA}(s,m,a) \triangleq p_{SM}(s,m)P(a|s,m) = p_{SM}(s,m)\pi_B(a|s,m)$$

$$p_A(a) \triangleq \sum_{(s,m) \in \mathcal{M}} p_{SMA}(s,m,a)$$

$$p_{MA}(m,a) \triangleq \sum_{s \in \mathcal{M}} p_{SMA}(s,m,a)$$

$$p_{SA}(s,a) \triangleq \sum_{m \in \mathcal{M}} p_{SMA}(s,m,a)$$

From this we can define the monitoring metrics:

- *Mean Entropy:*

$$\bar{H}(\pi) = \frac{1}{|\mathcal{M}|} \sum_{(s,m) \in \mathcal{M}} \left[ -\sum_{a \in \mathcal{A}} \pi(a|s,m) \log \pi(a|s,m) \right]$$

- *Mutual Information between messages and actions*

$$I_m = \sum_{m \in \mathcal{V}} \sum_{a \in \mathcal{A}} p_{MA}(m,a) \log \frac{p_{MA}(m,a)}{p_A(a)p_M(m)}$$

- *Mutual Information between states and actions*

$$I_s = \sum_{s \in \mathcal{M}} \sum_{a \in \mathcal{A}} p_{SA}(s,a) \log \frac{p_{SA}(s,a)}{p_A(a)p_S(s)}$$

**Analysis.**   Figure 12 displays the evolution of these metrics after each iteration as well as the evolution of the success rate (a). As indicated by Eq. (5), doing self-imitation learning results in a decay of the mean entropy (b). This decay is similar for ABIG and ABIG-no-intent. The most interesting result is provided by the evolution of the mutual information (c). For ABIG-no-intent, we see that $I_s$ and $I_m$ slowly increase with $I_s > I_m$ over all iterations. This indicates that the builder policy $\pi_B(a|s,m)$ relies more on states than on messages to compute the actions. In this scenario the builder, therefore, tends to ignore messages. On the other hand, $I_s$ and $I_m$ evolve differently for ABIG. Both metrics first increase with $I_s > I_m$ until they cross around iteration 25. Then $I_s$ starts decreasing and $I_m$ grows. This shows that ABIG results in a builder policy that strongly selects actions based on the messages it receives which is a desirable feature of emergent communication.

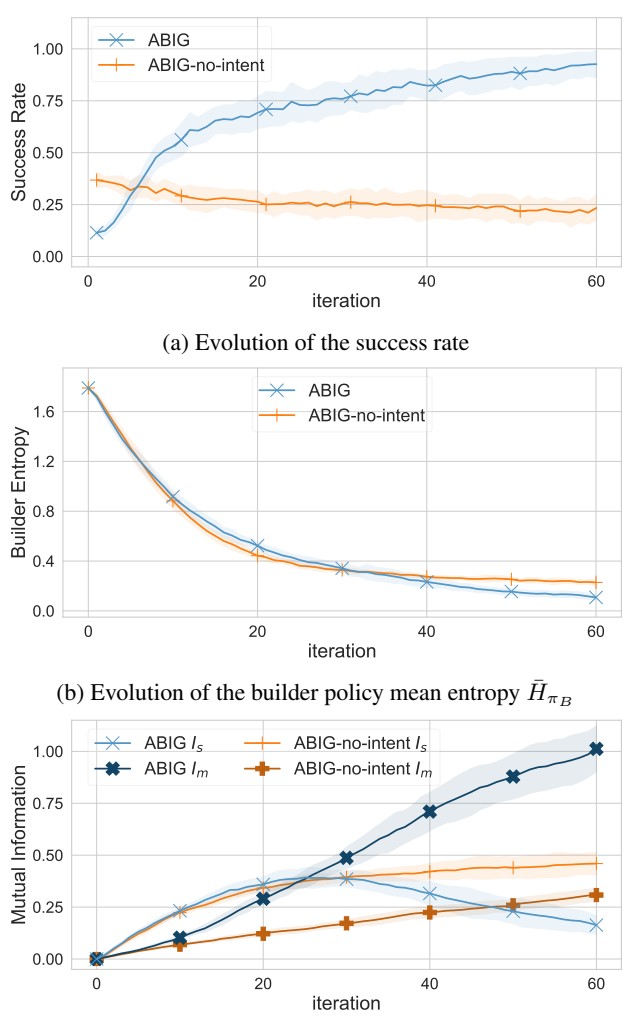

(a) Evolution of the success rate

(b) Evolution of the builder policy mean entropy $\bar{H}_{\pi_B}$

(c) Evolution of the mutual information $I_s$ and $I_m$

**Figure 12:** Comparison of the evolution of builder policy properties when applying ABIG and ABIG-no-intent on the 'place' task in BuildWorld. (a) ABIG enables much higher performance that ABIG-no-intent. (b) Both methods use self-imitation and thus reduce the entropy of the policy. (c) ABIG promotes the mutual information between messages and action which indicates successful communication protocols.

## B.2 ADDITIONAL BASELINE COMPARISON

We define two extra baselines:

- Stochastic: where the builder policy is a fixed softmax policy parameterized by a randomly initialized network;
- Deterministic: where the builder policy is a fixed argmax policy parameterized by a randomly initialized network.

In the performances reported in Figure 13, the architect has direct access to the exact policy of the builder ($\tilde{\pi}_B = \pi_B$) and uses it to plan and guide the builder during evaluation. We observe that the stochastic condition exhibits similar performances as the random builder. This indicates that, even if the architect tries to guide the builder, the stochastic policy is not controllable and performances are not improved. Finally, we would expect a deterministic policy to be more easily controllable by the architect. Yet, as pointed out in Figure 13, the initial deterministic policies lack flexibility and fail. This shows that the builder must iteratively evolve its policy in order to make it controllable.

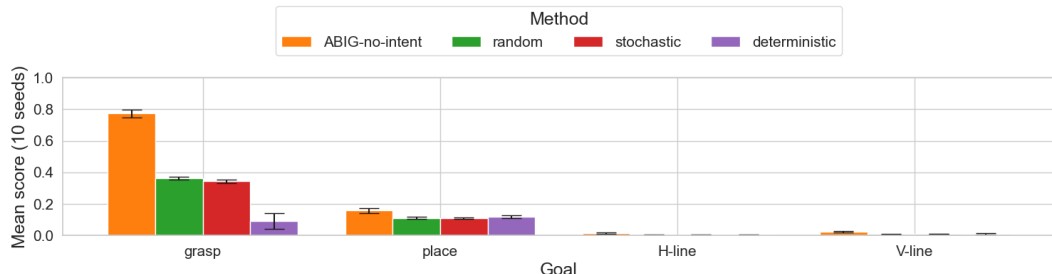

**Figure 13:** Baseline performance depending on the goal: stochastic policy behaves on par with random builder. Self-imitation with ABIG-no-intent remains the most controllable baseline.

### B.3 IMPACT OF THE VOCABULARY SIZE

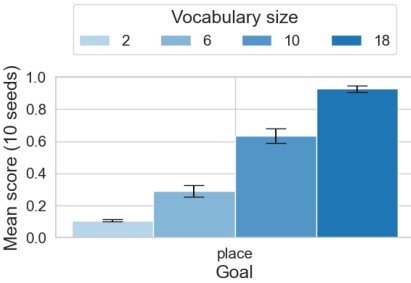

**Figure 14:** Influence of the Vocabulary size for ABIG on the 'place' task. Performance increases with the vocabulary size.

