# OpenReview forum: "Learning to Guide and to be Guided in the Architect-Builder Problem"
_ICLR.cc/2022/Conference — ICLR 2022 Poster_

### Official Review · Reviewer_BHGy · 2021-10-31

**Correctness:** 4
**Technical Novelty And Significance:** 2
**Empirical Novelty And Significance:** 2
**Recommendation:** 5
**Confidence:** 3

**Main Review:**

## Strengths
* An interesting setting and learning paradigm which seems to promote the emergence of generalizable communication protocols.

## Weaknesses
* Several assumptions seem to be made which limit the approach's direct applicability to more complex environments
  * Giving the architect access to the ground-truth environment model seems to be a very strong assumption.
  * The heuristic used in MCTS is not described in detail, but, presumably, it uses a significant amount of domain knowledge.
* It's unclear how the self-imitation learning works (see "Questions" below for more detail). My main concern is that this only works due to the fact that the architect has access to ground truth environment transition models such that it can exploit spurious correlations between messages and behavior in an untrained builder model. This seems unlikely to work at all even in this simple environment without access to the environment model.
* Lacking in experiments which attempt to understand the nature of the learned communication protocol. From section B.3 it seems like the agents simply learn to map messages directly to actions which is somewhat disappointing. If this is the case, then the generalization results may simply be down to the effectiveness of the MCTS procedure. In other words, the architect-builder pair learns messages that map to all possible actions, so the architect's job is reduced to standard MCTS with the original action space.

## Questions
* How is it possible for the builder to have any sort of coherent predictable behavior at the beginning of training? It seems this would be crucial for the self-imitation learning to work. In other words, an untrained builder agent will have no ability to interpret messages and modulate its behavior through them. As a result, the trajectories produced by the architect-builder pair will be no better than random and self-imitation of these trajectories shouldn't produce any meaningful results. It seems section 3.3 attempts to address this, but a more intuitive description would be appreciated.

**Summary Of The Paper:**

The authors propose a setting in which two agents with asymmetric information collaborate to complete a goal. They demonstrate that learning in this setting results in the emergence of communication protocols that generalize to new tasks.

**Summary Of The Review:**

This paper presents an interesting setting; however, it appears to rely too heavily on strong assumptions, and the communication protocols that emerge do not seem to exhibit any interesting properties (i.e. the communicating agent simply learns to output messages that correspond to the desired action for the acting agent).

---

> ### Author Response · Authors · 2021-11-12
> **ABP’s challenges lie in the interactions between the agents. This justifies prior knowledge about the physical world and the emergence of simple, yet successful, protocols.**
>
> We thank reviewer BHGy for their thoughtful review. Reviewer BHGy raised three concerns that we hope to clarify in the following three paragraphs. We have updated the manuscript accordingly.
>
> Reviewer BHGy found that giving the architect strong domain knowledge is a too restrictive assumption that could prevent our method from being transferred to more complex environments. We would like to emphasize that our proposed Architect-Builder Problem is a novel interactive learning setting that we specifically designed to analyze the underlying mechanisms required for two agents to negotiate a communication protocol while performing a task. In this context, our primary motivation is not to scale our approach to more complex tasks but rather to gain fundamental knowledge about this specific and original multi-agent problem. As a result, we believe that providing the architect with information about (1) the environment (transition function) and (2) the tasks (in the MCTS heuristic) is a sound assumption since it does not affect the interactions between the agents. Indeed, these considerations are related to the architect's proficiency at the task, and, as stated in the introduction, the architect is aware of the task and knows how to perform it: its challenge is to guide (or “teach”) the builder without a predefined communication protocol. Therefore, there should be no need for the architect to learn the physics of moving blocks from scratch (i.e. the environment transition function) or to figure out how to arrange blocks into structures (the MCTS heuristic). It can be regarded as prior knowledge as is the case with human architects (as noted by reviewer hMeT).
>
> Reviewer BHGy's found our intuitive explanation of ABIG not clear enough. We apologize for this lack of clarity and provide further explanations here. First, reviewer BHGy asks whether MCTS only exploits spurious correlations between messages and behaviors in an untrained builder model. Interestingly, the same question motivated our ablations choices. The ABIG-no-intent, deterministic and stochastic baselines are equivalent to three different fixed builder models. Their poor performances (particularly on harder tasks, provided in Figure 13) indicate that it is not sufficient to exploit spurious correlations between messages and behaviors to solve the Architect-Builder Problem. More elaborated mechanisms are required and, as developed in our intuitive explanation in section (3.3), ABIG relies on two mechanisms (reinforcement and forgetting) to create and reinforce action-messages associations that lead to success.  Second, reviewer BHGy wonders how the builder can have any sort of coherent predictable behavior at the beginning of training. This point is also examined in section 3.3 in which we analyze the dynamics that lead the builder policy to be "controllable".  Finally, figures 9 and 11(b) show that ABIG successfully updates the builder’s policy to make it controllable even if initial conditions are non-favorable. We hope that the pointers provided in this paragraph clarify the reviewer’s concerns and we would be glad to answer any remaining specific question on ABIG or self-imitation.
>
> Reviewer BHGy's last concern regards the lack of experiments to analyze the nature of the learned protocol. As mentioned in the last paragraph of Discussion and Future Work, we agree that this is an interesting research direction to pursue. However, as stated earlier, we decided to narrow the scope of this paper to the study of the underlying learning mechanisms of communication emergence, not of the emerged protocols themselves. Still, the properties that we monitor for this study such as the increase of the mutual information between messages and actions provided in figure 12, would be an interesting initial insight into the protocols’ properties. Finally, as noted by reviewer BHGy's, the learned protocol appears to be a mapping between messages and actions. We do not find this result disappointing. On the contrary, provided that the builder policy is conditioned on both states and messages, finding such a simple protocol in the space of all possible protocols is quite a success in itself. Additionally, the CoCo experiment that we use to derive this artificial setting demonstrates that humans succeed at it only if they emerge simple enough protocols. As a consequence, the protocol emerged by humans (positive or negative feedback) is arguably even simpler than the one derived by artificial agents.
>
> Important note: Figure references are given for the latest version of the manuscript uploaded on openreview.

---

> > ### Comment · Reviewer_BHGy · 2021-11-18
> > **Reply**
> >
> > Thank you for the added context. After reading the reply and some thought, I believe I understand how the self-imitation approach can result in a coherent message-conditioned policy. Some concerns remain.
> >
> > With respect to the scalability to more complex domains and/or assuming less domain knowledge: I don't believe that either of those things are required for a method to be worthwhile; however, in their absence I expect that the proposed method would learn a communication protocol that is more interesting than a simple mapping between communication symbols and low-level builder actions. While the authors consider this a success in and of itself, prior work has shown that standard MARL approaches that make, in my opinion, less restrictive/strong assumptions, are able to learn similar protocols. (e.g. Cooperative Communication environment in Lowe et al. 2017 [1]). Without a rich communication protocol, the approach appears to be MCTS with an extra step (self-imitation) for which the utility seems unclear. It would perhaps be useful if the authors could provide a real-world example for where this particular interactive learning setting would be applicable. The authors write that their aim is to "gain fundamental knowledge about this specific and original multi-agent problem." However, it is not clear to me yet what makes this problem inherently worth studying. As such, I must evaluate the work based on the outcomes of the proposed method, which do not appear to be significant in comparison to existing results in similar settings when taking into account the amount of assumed information.
> >
> >
> > [1] Lowe, Ryan, et al. "Multi-Agent Actor-Critic for Mixed Cooperative-Competitive Environments." Advances in Neural Information Processing Systems 30 (2017): 6379-6390.

---

> > > ### Author Response · Authors · 2021-11-18
> > > **Specific remarks**
> > >
> > >
> > > “the approach appears to be MCTS with an extra step (self-imitation)”: as we discuss in the paper with the ablation studies (no-intent, stochastic and deterministic) the solution is more complex than that and the co-evolution of the builder’s policy through guiding interactions (motivated by the shared intent prior) is essential.
> > >
> > > “the outcomes of the proposed method, which do not appear to be significant in comparison to existing results in similar settings”: in the light of our clarification of the difference between ABP and MARL, are there still “similar settings” that come to mind? If yes, would it be possible to provide references?
> > >
> > > “however, in their absence [domain knowledge] I expect that the proposed method would learn a communication protocol that is more interesting than a simple mapping between communication symbols and low-level builder actions”: would it be possible to develop on this intuition that less domain knowledge yields more complex communication protocols (if possible, specifically in the ABP setting where rewards are unavailable to the builder)?
> > >
> > > [Could we ] “provide a real-world example for where this particular interactive learning setting would be applicable.”: Multiple real-world examples where ABP could be the modeling framework are discussed in point (3) above. Note that our work does not target a specific robotic task in particular. Rather, and in the same vein as [3], but in the machine learning context, this line of inquiry proposes a setting to investigate the underlying learning mechanisms required to flexibly evolve communication protocols.
> > >
> > > [3] Vollmer, A.L., Grizou, J., Lopes, M., Rohlfing, K. and Oudeyer, P.Y., 2014, October. Studying the co-construction of interaction protocols in collaborative tasks with humans. In 4th International Conference on Development and Learning and on Epigenetic Robotics (pp. 208-215). IEEE.

---

> > > ### Author Response · Authors · 2021-11-18
> > > **The ABP setting should not be confused with MARL. We discuss why it is a more realistic setting for the investigation of the emergence of communication when an agent ignores the task at hand.**
> > >
> > > The authors thank the reviewer BGHy for the reply and are glad that their response dissipated the confusion about self-imitation learning.
> > >
> > > From the reviewer’s response, it seems that an important confusion remains about the learning setting we propose. Indeed, reviewer BGHy seems to believe that ABP is (1) quite similar to MARL (2) has more restrictive assumptions, and (3) potentially not inherently worth studying.
> > >
> > > (1) **ABP is inherently different from MARL:** As discussed in the introduction and in the related work section (and noted by reviewer hMeT), ABP is fundamentally different from MARL settings that consider Markov Games [1] in which “each agent also has an associated reward function” such as [2]. Indeed, in ABP the builder does not have access to a reward signal. The absence of such explicit supervision makes ABP a more challenging setting than classic MARL since the rewardless agent must rely on weaker learning signals derived from high-level priors.
> > >
> > > (2) **ABP lets go of the shared reward assumption:** We believe that providing all the agents with a shared reward such as done in common MARL work is restrictive and unfeasible for autonomous emergence of communication where some agents ignore the task at hand. For example, consider the cooperative communication task of [2] where a speaker that observes a target landmark must learn to communicate it to a listener that ignores which landmark it must reach. In such a setting, how can one assume that the listener receives, at each timestep, a reward based on its distance to a target landmark it ignores? It appears difficult to explain where this external reward could come from in a real-world setting. Therefore, our proposed ABP setting, where an agent cannot compute a reward based on the information it ignores, is more representative of such real-world autonomous emergence of communication.
> > >
> > > (3) **ABP is worth studying:** In light of this, we believe that the ABP connects with several other research problems. First, ABP could help design HRI systems where agents learn only through interactions instead of relying on pre-defined rewards or demonstrations. Then, the inherent constraints of ABP are reminiscent of the way children learn through interactions with their peers, an investigation at the core of developmental psychology. Finally, as mentioned in the introduction of our paper, ABP can be studied through the lens of computational semiotics to understand the fundamental mechanisms of communication emergence (such as in the CoCo game experiment of [3]). In all of these real-world applications, ABP provides a more realistic setting than classic MARL.
> > >
> > > Finally, we beg to differ with reviewer BGHy’s evaluation criteria: “I must evaluate the work based on the outcomes of the proposed method”. We believe that the interest and impact of our work should not be restricted to the apparent simplicity of the learned communication protocol. Indeed, this would overlook our work’s true contributions: a novel learning setting for communication emergence, an algorithmic solution based on well-motivated high-level priors, a thorough analysis of the resulting learning mechanisms, and a comprehensive evaluation of their generalization capability.
> > >
> > > [1] Littman, M.L., 1994. Markov games as a framework for multi-agent reinforcement learning. In Machine learning proceedings 1994 (pp. 157-163). Morgan Kaufmann.
> > >
> > > [2] Lowe, Ryan, et al. "Multi-Agent Actor-Critic for Mixed Cooperative-Competitive Environments." Advances in Neural Information Processing Systems 30 (2017): 6379-6390
> > >
> > > [3] Vollmer, A.L., Grizou, J., Lopes, M., Rohlfing, K. and Oudeyer, P.Y., 2014, October. Studying the co-construction of interaction protocols in collaborative tasks with humans. In 4th International Conference on Development and Learning and on Epigenetic Robotics (pp. 208-215). IEEE.

---

> > > > ### Author Response · Authors · 2021-11-19
> > > > **Additional diagram to illustrate ABP vs. MARL**
> > > >
> > > > We have added a figure to highlight the contrasts between our ABP setting and the MARL setting (Figure 8 of the Supplementary Material).
> > > >
> > > > We remain available to discuss any remaining concerns.

---

> > > > ### Comment · Reviewer_BHGy · 2021-11-19
> > > > **Unconvinced of ABP setting's assumptions and whether they are realistic**
> > > >
> > > > It is clear that this setting differs from MARL, but the question is whether those differences are grounded in more realistic assumptions. The authors appear to agree with this sentiment, given points 2 and 3. It's not clear that the shared reward assumption is a particularly strong or unrealistic one, especially in a setting that allows communication. What is stopping the architect from simply communicating the reward it observes to the builder agent? This would be analogous to a parent providing instructions to their child on how to play with a toy and providing verbal affirmation when the child does it correctly. What real-world setting exists where a listener agent is motivated to modulate their behavior by communication signals received by a speaker agent, the speaker agent has an objective in mind, but for some reason the speaker cannot simply just communicate their knowledge of this objective to the listener?
> > > >
> > > > After reading the responses to myself and other reviewers, I do feel that I more clearly understand the intention of the work, which is to learn a communication protocol between an architect and builder in a setting that restricts interactions between the two. In this case, it is understandable that prior knowledge is provided to the architect if we assume that the architect may be a stand-in for a human or a pre-trained artificial agent with more knowledge than the builder; however, when the emphasis is placed on the restriction of interactions, I think it is important to scrutinize the choices of what exactly is being restricted. The argument of the authors appears to be that the restrictions of the ABP setting (no shared rewards) are of significant real-world interest, and the emergence of any useful communication protocol in this setting is noteworthy. As stated above, the real-world applicability of the setting is unclear. Even so, I would have a more positive impression of the paper if such a setting induced surprising results (e.g. a rich communication protocol), but that does not appear to be the case. I will keep my current score.

---

> > > > > ### Author Response · Authors · 2021-11-20
> > > > > **Manipulating the reward of an agent is not communication.**
> > > > >
> > > > > The authors are grateful for reviewer BHGy's fast response. They are glad that the discussion clarified the intention of this paper and showed the soundness of its associated assumptions. We also thank the reviewer for developing their concerns which enabled us to identify one key point of disagreement.
> > > > >
> > > > > From the point of view of the authors, **communication can only happen in the observation space and in no way in the reward space.** To set the reward of another agent would require manipulating another agent’s learning algorithm (to have access and the possibility to modify its “mind/inner workings”). This is unfeasible in a decentralized multi-agent setting.
> > > > >
> > > > > In particular, regarding reviewer BHGy’s remark “What is stopping the architect from simply communicating the reward it observes to the builder agent?”: the architect sends messages that are **observed** by the builder, therefore messages are observations, not rewards. More colloquially, in the CoCo game setting, the architect is displaying arbitrary shapes to communicate with the builder, not opening the builder’s software and tweaking its reward.
> > > > > In the situation where the architect would communicate a scalar reward to the builder, the builder would then receive an additional scalar observation (the communicated reward). But, in this situation, **how would the builder know how to interpret this scalar observation and use it as a reward** (i.e. integrate it into its learning algorithm and use it as a learning signal)? For all of what the builder knows, the architect could be communicating anything: the room temperature, its battery levels, or the number of dollars it has at the bank...
> > > > >
> > > > > Similarly in the situation of “a parent providing instructions to their child on how to play with a toy and providing verbal affirmation when the child does it correctly.” the parent explicitly influences observations, not rewards. Indeed, the child **hears** the parent’s voice and internally processes it into words to form meanings. The child, then, extracts an **intrinsic reward** to judge how good it is doing and what it should do. But in no way is the parent **directly releasing dopamine in the child's brain** to manipulate its reward pathway.  The former is what the ABP setting aims to investigate: how can one develop a communication channel (i.e. a way of influencing another agent’s observations) to guide another agent toward a task it ignores. While the latter would be using FRL to tackle this problem.
> > > > >
> > > > > To the question “What real-world setting exists where a listener agent is motivated to modulate their behavior by communication signals received by a speaker agent, the speaker agent has an objective in mind, but for some reason, the speaker cannot simply just communicate their knowledge of this objective to the listener?”. We believe that the “parent teaching a child” is precisely a good example. Indeed, pre-linguistic children don’t know exactly how to interpret their parents’ instructions/encouragements. They must figure it out while their parents identify how instructions are perceived and adapt them accordingly. In addition, we have detailed in our last comment a number of real-world cases in AI, HRI, DevSci, and experimental semiotics (see for example the CoCo game video [here](https://www.youtube.com/watch?v=TAeURLIpiEo&ab_channel=InriaFlowers)). If these real-world examples do not convince the reviewer, could they please clearly explain why they think that such examples are not valid?
> > > > >
> > > > > Finally, regarding “I would have a more positive impression of the paper if such a setting induced surprising results (e.g. a rich communication protocol), but that does not appear to be the case.” As addressed in the previous response: the authors believe that **the real success is to emerge any functional communication protocol, regardless of its complexity.** In [1], human participants to the CoCo game only manage to emerge a simplistic “positive/negative feedback” communication protocol, yet, humans are arguably the most successful thing we know of at communication and we should rejoice that our AIs succeed at similar experiments.
> > > > >
> > > > > The authors hope that this clarifies why ABP is grounded in more realistic assumptions than MARL or FRL when it comes to studying the mechanisms underlying the emergence of communication where an agent is unaware of the task at hand. Additionally, authors hope that, with the communication constraint clarified, the success of being able to solve the task at hand by evolving a simple communication protocol appears more vividly.
> > > > >
> > > > > [1] Vollmer, A.L., Grizou, J., Lopes, M., Rohlfing, K. and Oudeyer, P.Y., 2014, October. Studying the co-construction of interaction protocols in collaborative tasks with humans. In 4th International Conference on Development and Learning and on Epigenetic Robotics (pp. 208-215). IEEE.

---

> > > > > > ### Comment · Reviewer_BHGy · 2021-11-21
> > > > > > **Thank you**
> > > > > >
> > > > > > I appreciate the further clarification provided by the authors, and I have certainly reframed my evaluation of this work given the added context.
> > > > > >
> > > > > > I recognize that communicating rewards is conceptually distinct from communicating observations and requires further assumptions regarding the nature of the builder's learning procedure. I have some thoughts on this point:
> > > > > > The method proposed by this paper still requires assumptions on the nature of the builder agent. In particular, it requires that the builder agent learns through self-imitation and is capable of receiving communication from the architect. If we are to be making any assumptions regarding the nature of the builder's learning algorithm, why do we stop at observations? It is true that we cannot directly control dopamine pathways in a human's brain; however, this seems to be a case of attempting to recreate a human-like learning setting when it may not be prudent to do so. This is where I believe the fundamental tension lies. The authors appear to have created a setting that is constrained in the ways that humans without shared language must learn to communicate. I do believe that this is inherently interesting; however, whether or not it is of interest to the ICLR community hinges on whether the method presented is useful in advancing the state of AI. In this respect, I don't believe the presented work offers enough. I can't imagine a scenario in which we want to train an artificial agent, are able to make the assumptions that it learns via self-imitation and that we can send messages to it, but cannot simply send it a scalar reward (no more complicated than sending communication messages) and have it learn from that signal instead of or in addition to self-imitation. It's not clear at all that these are stronger assumptions than those made in the paper, especially when we consider the fact that we are training artificial agents, not humans.

---

> > > > > > > ### Author Response · Authors · 2021-11-24
> > > > > > > **Thank you for the discussion. We believe there exist important real-world AI applications.**
> > > > > > >
> > > > > > > The authors thank the reviewer for the continued discussion. We are glad that we managed to clarify the scope of our work and address the initial concerns that were raised. We appreciate that this discussion enabled the reviewer to *“reframe their evaluation of our work”* and realize that it is *“inherently interesting”*. We understand that now the reserve of the reviewer lies with the bigger picture of our work and *“whether or not it is of interest to the ICLR community hinges on whether the method presented is useful in advancing the state of AI”*. We wish to highlight in the paragraphs below that, even if our setting is devised to theoretically and experimentally investigate the fundamental learning mechanisms of the emergence of communication, it also presents practical interests in advancing the current state of AI. Indeed, it proposes a setting for multi-agent interactions where fewer assumptions are required than in a setting where agents communicate rewards. Namely, there are few assumptions about the available communication channel or about an agent’s awareness of the other agent’s way of interpreting messages.
> > > > > > >
> > > > > > > Specifically, in the ABP setting, there are no assumptions made by the architect or the builder about the other agent's way of interpreting messages and there is no restriction on the communication medium (sounds, images, scalars, etc.). Concretely, in our solution, the architect does not need to assume anything about the builder (it does not have to know its learning algorithm, i.e. self-imitation in our case) and the builder only has to assume that the architect is trying to optimize an objective (shared intent) that may remain unknown to the builder.
> > > > > > >
> > > > > > > This makes ABP a relevant modeling setting for contemporary real-world applications such as the ones that tackle the calibration problem in Brain-Computer Interfaces (BCI) where a user must use Electroencephalography (EEG) signals to guide a prosthetic arm (“Calibration-Free BCI Based Control”, Grizou et al., 2014, AAAI; “Interaction-Grounded Learning”, Xie et al., 2021, ICML). We will discuss these potential applications in the camera-ready version of the paper.
> > > > > > >
> > > > > > > That being said, we understand that now the discussion is drifting towards personal beliefs about how to best advance the state of AI and the authors do not feel that it is their place to argue. We acknowledge the reviewer’s opinion and thank them for the valuable discussion.

---

### Official Review · Reviewer_hMeT · 2021-11-02

**Correctness:** 4
**Technical Novelty And Significance:** 4
**Empirical Novelty And Significance:** 4
**Recommendation:** 8
**Confidence:** 4

**Main Review:**

Overall, I find the setting and the proposed algorithm novel and interesting. I really like that authors carefully distinguish their setting from RL and IL, and give very nice intuitions about the learning dynamics of the proposed algorithm. Although the architect-builder relationship has been explored in previous work, preventing the builder from knowing the rewards is, to my best knowledge, not explored. However, I feel like this setting makes more sense if the architect were a human, as they may have implicit intent that the builder has to manage to interpret. I'd encourage the authors to give a concrete example where this setting is useful in the agent-agent setting. In general, what is the motivation of this work? Is it a computational model for studying human-human communication? Or is it offering a learning framework that aim for specific practical scenarios? The motivation should be stated more clear in the introduction.

Another suggestion is to rewrite the problem formulation using the MDP formalism (i.e. <S, A, T, R, gamma>), clearly defining the MDPs and policies of the architect and the builder. In addition, the description of the interaction needs to be more precise: right now, it does not specify whether the architect sends a message to the builder after every step or the builder can take multiple steps after a message.

The main limitations of this work are (1) the architecture requires access to a simulator of the environment (which may not be a problem for a human) and (2) the simplicity of the experimented environment. The authors adequately acknowledge these limitations.

Suggested related work (on more complex environments):

[1] Collaborative Dialogue in Minecraft. https://aclanthology.org/P19-1537.pdf

[2] Hierarchical Decision Making by Generating and Following Natural Language Instructions. https://arxiv.org/pdf/1906.00744.pdf

[3] Interactive Learning from Activity Description. https://arxiv.org/pdf/2102.07024.pdf

[4] Neural Abstructions: Abstractions that Support Construction for Grounded Language Learning. https://arxiv.org/pdf/2107.09285.pdf

[5] Incorporating Pragmatic Reasoning Communication into Emergent Language. https://arxiv.org/pdf/2006.04109.pdf

**Summary Of The Paper:**


**After rebuttal**: I am keeping my current score. I am positive about this framework as it presents a better model for multi-agent communication, especially enriching the communication among agents over the fixed, restricted reward-based communication protocol in traditional RL.

**Before rebuttal**: The paper proposes an architect-builder problem setting where the architecture guides the builder to accomplish a goal by sending messages. Only the architecture knows the goal and has access to rewards, whereas only the builder can act in the environment. This setting is distinct from traditional reinforcement learning and imitation learning.

Drawing inspiration from cognitive science theories, the authors devise an algorithm for learning a communication protocol between the architect and the builder. On grid-world tasks, they show that learned the communication protocol can generalize to previously unseen tasks.


**Summary Of The Review:**

The paper presents a novel and interesting setting and algorithm. They nicely implement ideas from cognitive science in an MDP setting. Even though the formulation needs to be more rigorous and the experiment environments are simplistic, I think the contributions are interesting enough to draw attention of the community in the future. I recommend acceptance.

---

> ### Author Response · Authors · 2021-11-12
> **Thank you for the review. ABP is indeed a computational model to study agent-agent communication.**
>
> The authors would like to thank reviewer hMeT for highlighting this work’s novelty and attentive comparison to related work. In addition, the authors have incorporated the valuable suggestions regarding the problem definition and the MDP formulation.
>
> The motivation for investigating the agent-agent setting rather than the more classical human-agent setting comes from Experimental Semiotics. As pointed out in the related work section, standard HRI formulations (Grizou et al., Cederborg et al.) are restricted in the sense that they consider static humans (architects), meaning that they follow a predefined communication protocol without adapting their behavior to the robot’s reaction. This might present an applicative interest (for example humans teaching robots) but greatly restricts the analysis of interactive emergence of protocols. In contrast, an agent-agent formulation as the one we propose enables both agents to learn and adapt. This resembles more closely to the common Experimental Semiotic setting used to investigate the fundamental learning and teaching mechanisms that manifest in the absence of predefined communication protocols. Finally, it should be noted that the proposed setting is not restricted to AI-AI interactions but rather proposes a flexible formulation that can handle humans, AIs, or even models of human behavior. Consequently, an agent-agent formulation fits best the scope of this paper that is to investigate the underlying mechanisms by which multiple agents evolve communication protocols to cooperate. The introduction and related work have been updated to highlight this motivation.
>
> We hope that this answers reviewer hMeT’s concerns and we remain available for further discussion.

---

> > ### Comment · Reviewer_hMeT · 2021-11-19
> > **Thanks for your response!**
> >
> > Thanks for your response. The agent-agent setup makes sense if the goal is to study emergent communication. I do hope that the paper would formalize the MDPs more rigorously and focus on describing the novelty of the communication protocol (even with a figure that draws the contrasts among this protocol, the RL protocol, and the IRL protocol). From reading other reviews, I can see that it might be hard for someone who works on traditional RL to see the distinction from just reading the text.

---

> > > ### Author Response · Authors · 2021-11-19
> > > **Additional diagram and MDP definition**
> > >
> > > The authors thank the reviewer for the constructive feedback.
> > >
> > > We have updated the PDF manuscript by adding a figure to highlight the contrasts between our ABP setting, the MARL setting, and the IRL setting (Figure 8 of the Supplementary Material). We have also extended the MDPs definitions in Section 3.1.
> > >
> > > Please don't hesitate if you have additional questions or suggestions.

---

### Official Review · Reviewer_FMPZ · 2021-11-02

**Correctness:** 3
**Technical Novelty And Significance:** 3
**Empirical Novelty And Significance:** 3
**Recommendation:** 6
**Confidence:** 4

**Main Review:**

The modelling of the problem is sensible (i.e. using two distinct, connected, MDPs). It would support clarity if this design decision was explained/backed with respect to established research on multi-agent modelling. Are any reductions possible? It seems that several challenges of the proposed framework have been solved in different MAS works, so it would be userful to be clear here.

The goal of the paper is relevant for the conference, as a communication channel is directly learned and the framework somewhat works towards the bigger goal of AGI. However, it would be useful to add more real world applications of the approach. What would intermediate applications be before a perfect version of the approach is available?

The related work is informative, but I am missing comparison to recent works on such learning protocols - e.g. [1]. There are differences between the frameworks and [1] might work towards HRI (as also mentioned in the paper), but the direction seems quite related. It would be also interesting to dwell on the reusability of the proposed algorithm for the problem at hand.

The authors themselves note in the discussion of the paper that the employed methods are suboptimal for the defined learning problem (i.e. using behavioral cloning and MCTS). It would be beneficial to clearly state the learning challenges when defining the protocol, as it is a main part of the contribution.

I am also wondering to what extent it might be possible to reuse established approaches for emerging communication, as already cited in the paper (e.g. [2]).

The empirical evaluation is informative, although the random building agent does not give too much additional insight. It would be good to show relevant ablations in the main part of the paper.

A minor comment with respect to the figure of the approach: I find the pseudocode of the algorithm quite helpful (in my opinion, more helpful), but it is in the suppl. Material.

[1] Nguyen, K., Misra, D., Schapire, R., Dudík, M. and Shafto, P., 2021. Interactive Learning from Activity Description. arXiv preprint arXiv:2102.07024.
[2] Foerster, J.N., Assael, Y.M., De Freitas, N. and Whiteson, S., 2016. Learning to communicate with deep multi-agent reinforcement learning. arXiv preprint arXiv:1605.06676.


**Summary Of The Paper:**

The paper proposes an approach for interactive learning between a so-called architect and builder agent, which is a different, but related protocol to RL or imitation learning. Under the assumption that the architect knows the target dynamics and reward function, the authors focus on learning the communication between the two parties, such that the builder can solve the MDP.

After formalizing the setting in the multi-agent paradigm, where two individual MDPs are defined, the authors propose an algorithm with so-called modelling- and guiding phases. In each phase, architect and builder gather datasets, from which policies are extracted via behaviour cloning and planning.

The paper is evaluated on a proposed block environment for the problem and uses a random building agent and combination of random modelling phase with proposed guiding phase as baselines. The results show that the proposed approach is superior wrt the baselines for solving the block environment. In addition, experiments show that the learned communication channel can potentially be reused for solving other tasks.

**Summary Of The Review:**

The paper proposes an interesting and relevant framework for interactive learning / teaching between two agents. The model choice and solution is sensible, and the empirical evaluation can shows that the initial approach for the problem works sufficiently well for the toy problem. The learning challenges with respect to other works in MAS could be clarified in more detail.

---

> ### Author Response · Authors · 2021-11-12
> **Yes ABP is a multi-agent problem but its novelty and challenge lies in the fact the builder does not have access to a reward**
>
> We thank reviewer FMPZ for their thorough review and constructive remarks.
>
> ### Main response
>
> Reviewer FMPZ concerns relate to the connection between the Architect-Builder Problem (ABP) and traditional Multi-Agent setups. In what follows, we clearly expose the similarities and differences as well as the specific challenges. We have updated the manuscript accordingly.
>
> As we state in the paper, ABP is a multi-agent learning problem and relates to multi-agent RL (for example our proposed MDP decomposition is inspired by factored-MDPs [1,2]). Therefore, ABP shares the usual challenges of multi-agent learning such as non-stationarity. This latter is often dealt with by considering the centralized training, decentralized execution paradigm [3], yet in our case, we consider the more challenging fully decentralized training setting and we rely on interaction frames instead.
>
> What makes the novelty of ABP as a multi-agent learning setting is, as noted by reviewer hMeT, the absence of reward signal available to the builder. Therefore, the builder has to learn the meanings of messages (alongside performing the task at hand) without relying on a learning signal provided by a reward. This particular point sets our setting apart from works like [4] and prevents us from relying on similar methods.
>
> ### Additional Response
>
> We thank reviewer FMPZ for the pointer to ILLIAD [5]. We now discuss this work in the last paragraph of our related work section.
>
> ### References
>
> [1] Boutilier, C., 1999. Sequential optimality and coordination in multiagent systems. IJCAI .
>
> [2] Guestrin, C., Koller, D., Parr, R., 2001. Multiagent Planning with Factored MDPs. NeurIPS.
>
> [3] Lowe, R., Wu, Y., Tamar, A., Harb J., Abbeel, P., Mordatch, I., 2017. Multi-Agent Actor-Critic for Mixed Cooperative-Competitive Environments. NeurIPS.
>
> [4] Foerster, J.N., Assael, Y.M., De Freitas, N. and Whiteson, S., 2016. Learning to communicate with deep multi-agent reinforcement learning. NeurIPS.
>
> [5] Nguyen, K., Misra, D., Schapire, R., Dudík, M. and Shafto, P., 2021. Interactive Learning from Activity Description. ICML.

---

### Official Review · Reviewer_rTG5 · 2021-11-04

**Correctness:** 2
**Technical Novelty And Significance:** 2
**Empirical Novelty And Significance:** 2
**Recommendation:** 3
**Confidence:** 4

**Main Review:**

I believe that this is a well-written paper, with solid motivation, and thoughtful care in crafting the approach. However, the bulk of my criticism focuses on the assumptions underlying the approach, and it’s situation relative to prior work in hierarchical, and specifically Feudal RL, an entire body of work that is omitted in the paper.

Concretely, I  believe that providing the architect access to the transition function is an incredibly strong assumption that undercuts this work. Existing work in Hierarchical RL that learns separate high- and low-level policies, under similar assumptions as the Architecture-Builder setup (though possibly without the restriction of the discrete message channel) *do not* make this perfect transition assumption, and instead focus on alternative means of learning the “architect.” Some examples are HIRO (https://arxiv.org/abs/1805.08296), Feudal Networks (https://arxiv.org/abs/1703.01161), h-DQN (https://arxiv.org/abs/1604.06057), amongst many many more (there are many other Feudal RL approaches, including more modern ones that appear in the skill discovery literature).

This leads into my second key weakness, which is around the evaluation. I understand that full system evaluations are hard, but in this specific case, I think the current ablations trivially fail, and don’t provide much insight into the proposed approach. Instead, I would love to see other work that looks at more traditional HRL approaches, or that relaxes the some of the assumptions made in this work. Specifically, relaxing the “discrete communication channel” assumption would allow for out-of-the-box comparisons to HIRO and h-DQN, as well as more recent work. Other ablations I’d like to see in future versions of this paper would look at versions of the Architect learning without access to transition rewards.

Typos/Style/Questions:

Broad Positioning Question: I love the Architect-Builder problem statement, but it’s not immediately clear to me why we need to restrict the communication mechanism between the architect and the builder/what that gives us more broadly. If the goal is to see how we might emerge a “language” (discrete tokens, similar to work on emergent communication work) it’d be nice to see how well an Architect or Builder learned with this approach transfers to “real-language” settings (e.g., adapt to natural language instructions!). If the goal is broadly coordination, I definitely think falling back on the Feudal RL literature is important; looking at latent continuous representations of “intent” is pretty pervasive in HIRO and newer work such as VALOR (https://arxiv.org/abs/1807.10299).

**Summary Of The Paper:**

This paper presents an approach — Architect-Builder Iterated Guiding — a method that tackles what is presented as an “Architect-Builder” problem: a scenario in which two actors, an Architect, with knowledge of a high-level goal, or reward function, must communicate over a discrete channel with a Builder who can take actions in the environment based on the Architect’s message. In many ways this resembles a **hierarchical reinforcement learning** setup, specifically reminiscent of Feudal Reinforcement Learning (“Feudal Reinforcement Learning,” Dayan et. al. NeurIPS 1992). The paper presents a motivated, easy-to-understand algorithm for training both the Architect and the Builder, and evaluate on a series of “construction” based grid-world tasks (resembling GridLU, MiniGrid, or Mazebase).

Namely, the proposed approach takes inspiration from Experimental Semiotics and separates learning into separate **interaction frames** (similar again to multi-phase approaches in the Hierarchical RL literature, e.g. in HIRO — “Data Efficient Hierarchical Reinforcement Learning,” Nachum et. al. 2018; https://arxiv.org/abs/1703.01161). These interaction frames consist of a modeling frame, in which the architect learns a model of the builder after “rolling out” and sending messages/watching the builder’s actions, followed by a guiding frame where the architect exploits its model of the builder to produce the “optimal” actions via a heuristic driven Monte Carlo Tree Search (rather than explicitly estimate a value function, which is costly and noisy). Making this all possible is that the architect has full knowledge of the high-level reward, **in addition to the ground-truth state transition function**. This lets the architect explicitly search over messages to send to the builder (as it fully observes the builder’s rollout), then improve the builder by “self-imitating” over this new data, mirroring a bi-phase optimization setup.

The evaluation focuses on the proposed model, and two simple ablations: one where the architect has “no-intent” at training, sending random messages, and one in which the builder takes random actions. The paper also presents (in the main body and appendix) a meaningful, thoughtful intuitive explanation of the learning dynamics of the architecture and the builder — more papers should dedicate portions of the main body to explanations such as this!

**Summary Of The Review:**

In general, I believe this is a well-written, well-motivated paper that draws some great ideas from Experimental Semiotics. Furthermore, the discussion and analysis of the learning dynamics of the various components of the approach should be a mainstay of future systems driven work in ML.

However, I am deeply concerned with the assumptions made with this approach, stemming from the architect’s access to the ground-truth state transition function. This coupled with a missing discussion of related work in hierarchical RL and Feudal RL specifically, and a more thorough evaluation including comparison to these methods and other relevant ablations inform my decision to lean towards rejecting this paper.

---

> ### Author Response · Authors · 2021-11-11
> **This work investigates the learning mechanisms involved when multiple agents collectively evolve a communication protocol**
>
> We thank reviewer rTG5 for their review, their valuable feedback, and for highlighting the quality of our explanation of the learning dynamics.  Our understanding is that rTG5’s concerns can be summarized as follows:
>
> 1. giving the architect access to the environment transition function is a restrictive assumption,
>
> 2. the relationship between this method and Feudal RL (FRL) is not clear, and
>
> 3. a comparison with FRL methods on the proposed tasks is missing.
>
> We believe that these concerns might stem from a misunderstanding about our motivation and goals, which we aim to clarify below. We have additionally updated our manuscript to further emphasize these points and added the related references.
>
> In the vein of Experimental Semiotics, our work aims to deepen our understanding of the fundamental learning mechanisms involved in the emergence of communication protocols. To address this question, we propose and investigate a formulation of the CoConstruction Game for artificial agents.
>
> 1. Figure 1 has been updated to better illustrate the co-construction experimental setting that inspired the Architect-Builder Problem (ABP). Specifically, the challenge is to evolve a mutually understood communication protocol that can be used by the architect to guide the builder. To do so, the architect must model the evolving builder’s reactions and determine how to influence them. Thus, when investigating the emergence of communication protocols, forcing the architect to learn the physics of stacking blocks from scratch (i.e., the environment’s transition function) is both of little interest and orthogonal to our primary investigation. Moreover, as noted by reviewer hMeT, human participants possess such prior world models. Consequently, assuming that the architect has access to the transitions and reward models remains a reasonable assumption and doesn’t undermine the impact of our work.
>
> 2. As discussed (see Related Work, 2nd paragraph), although the proposed ABP setting implements hierarchies between agents it remains fundamentally different from the Hierarchical Learning settings tackled by HIRO, h-DQN, FuN, or Valor. Indeed, FRL deals with single-agent learning and improves learning performance by enforcing a hierarchy in an agent’s learning algorithm. Thus, the definition of the hierarchy, and in particular the communication protocol between the different modules of an agent’s algorithm, are design choices. Often, messages are goal-states and their “meanings” are defined by intrinsic rewards that penalize the distance to the goal. In contrast, ABP investigates the emergence of communication in fully decentralized multi-agent learning. The hierarchy is therefore between different agents relying on independent learning algorithms. In this setting, it would be unfeasible to assume that one agent in the hierarchy has prior knowledge of the other agent’s learning algorithm (or that two agents that have never met before would share an existing communication protocol). Crucially, the builder has no explicit reinforcement signal, which makes ABP a novel learning setting (as noted by reviewer hMeT).
>
> 3. For these reasons, comparing ABP to FRL would violate the assumptions of the ABP and defeat its purpose of evolving a communication protocol. FRL is about leveraging pre-defined communication protocols whereas ABP is about emerging such communication protocols.
>
> We hope that this discussion clarifies the scope of our work and its differences with respect to Feudal Learning, eventually showing the soundness of the assumptions that were made. We would be glad to discuss any remaining or new questions that may arise from this discussion.

---

### Author Response · Authors · 2021-11-24
**A summary of the discussions at the attention of the AC (and the reviewers)**

We thank all reviewers for actively taking part in the discussion. We are glad that most of the reviewers' concerns enabled us to improve our paper. Unfortunately, there are still some concerns that remain unresolved even after extensive discussions. Therefore the authors wish to write this post to take a step back and summarize the current status of discussions and the improvements made to the manuscript. The authors tried to keep this summary concise but hope that it fairly reflects the reviewer’s views.

**Summary of the main concerns:**
* *“The fact that the architect has a priori environment knowledge is limiting”*. This point was raised by reviewers rTG5 and BHGy. Our argument is that this is a sound assumption that is supported by the fact that human architects possess such world models (Vollmer et al. 2014).
  * After discussion, reviewer BHGy agrees that *“it is understandable that prior knowledge is provided to the architect if we assume that the architect may be a stand-in for a human or a pre-trained artificial agent with more knowledge than the builder”*.
  * However, after several rounds of clarification, reviewer rTG5 stands by the opinion that *“The access the architect has to a known transition model is unjustifiable. [...] I believe it hurts the paper's greater goals of studying how good communication emerges.”*
* Reviewers found that the novelty of our approach was not clear.
  * Reviewer rTG5 stated that our proposed Architect-Builder Problem (ABP) is equivalent to Feudal/Hierarchical Reinforcement Learning. We addressed this concern by providing two arguments (unfortunately, these arguments do not seem to resonate with the reviewer):
      * Our setup includes two physically distinct agents with independent learning mechanisms.
      * Unlike HRL settings, the architect never directly influences the policy improvement of the builder (the builder is never given any intrinsic or extrinsic reward and relies on self-imitation to improve)
   * Reviewer BHGy and FMPZ asked for clarification about the differences between ABP and standard MARL approaches. We argued that the difference lies in the fact that, in ABP, the builder does not have an environmental reward function. Reviewer BHGy acknowledges the difference with MARL and notes that ABP is a relevant setting to model *“the ways that humans without shared language must learn to communicate”*. We are still waiting for an answer from reviewer FMPZ on this matter.
* Reviewers rTG5 and BHGy regretted the lack of analysis of the learned communication protocol. We highlighted that this work focuses on the learning mechanisms required to emerge protocols (not on the protocols themselves) and that we thoroughly investigate these learning mechanisms. Moreover, they found the emerging protocol disappointingly simple. We argued that the evolved instruction-giving protocol was simple but successful (and challenging to emerge given the constraints) and that humans also agree on simple protocols (Vollmer et al. 2014). If reviewer BHGy does not maintain this concern in their last response, reviewer rTG5 remains worried about it.
* Reviewers BHGy and FMPZ asked for concrete real-world applications. In our final response to reviewer BHGy, we give two references on Brain-Computer Interfaces where a user must use Electroencephalography (EEG) signals to guide a prosthetic arm. The authors will be glad to integrate these in the Discussion section of the camera-ready version of the manuscript.

**List of all modifications made to the manuscript:**
* Clarification of the motivations of ABP in the introduction
   * added motivations grounded in experimental semiotics (updated figure 1 with the human experiment that inspired ABP) as requested by reviewers BHGy and hMeT.
* Updated positioning
   * w.r.t  Inverse Reinforcement Learning and MARL (figure 8 in supplementary, added explanation in the introduction) as requested by reviewers hMeT and FMPZ.
   * Added supplementary related work to express the differences between ABP and F/HRL, as well as the difference between setting rewards and communication through observable channels (in related works and supplementary section A.5) as requested by reviewers rTG5 and BHGy.
   * Added discussion about related work in interactive learning (ILIAD) (last paragraph on section 4) as requested by reviewer FMPZ.
* Clarification of the problem definition:
    * Justification of architect’s access to environment transition function (section 2) as requested by reviewers rTG5 and BHGy.
    * Updated MPD formulation to make it more formal (section 3.1) as requested by reviewer hMeT.

---

### Decision · Program_Chairs · 2022-01-20

**Decision:**

Accept (Poster)

**Comment:**

This paper proposes a cognitive science-inspired interaction setting between two agents, an "architect" and "builder", in which the architect must produce messages to guide the builder to achieve a task. Unlike other related settings (such as typical approaches in MARL, HRL, or HRI), the builder does not have access to the architect's reward function, and must learn to interpret the architect's messages by assuming the architect is telling it to do something sensible. At the same time, the architect determines what is "sensible" by building a model of the builder's behavior and planning over it. This setting is common particularly in human-agent interactions, where humans may not be able to either (1) accurately communicate a scalar reward or (2) provide demonstrations, but can still provide information that the agent ought to be able to learn from. The paper demonstrates that the learned communication protocol generalizes well to new settings.

While this paper generated a lot of discussion, the reviewers did not come to a consensus on whether the paper should be accepted or rejected, with those in favor of the paper maintaining it should be accepted and those not in favor maintaining that it needs work. I have therefore done a particularly close read of both the paper and the discussion in order to weigh the pros and cons brought up by the reviewers.

The positive reviews clearly indicate that this work is insightful and of interest to researchers in the ICLR community (in fact, all reviewers mentioned they found the work interesting and well-written). In particular, Reviewer hMeT wrote: "I am positive about this framework as it presents a better model for multi-agent communication, especially enriching the communication among agents over the fixed, restricted reward-based communication protocol in traditional RL." I am inclined to agree with this assessment and find the communication setting studied in this paper to be much more ecologically valid for human-agent interaction settings than having humans communicate scalar rewards or provide demonstrations: humans are typically poor at the former and may not have the same embodiment to achieve the latter.

The negative reviews focused on a few cons: (1) the assumption that the architect has access to a ground-truth environment model, (2) confusion about differences from other related fields (e.g. feudal RL, MARL), and (3) lack of analysis of the communication protocol. I have considered these points, but do not feel any of them are fatal flaws: (1) From the perspective of human-agent interaction, I think it is very reasonable to assume that a human architect would have a good model of the world and would be generally proficient at solving tasks in the world. Making this approach work in the setting where the architect is *also* learning how the world works seems squarely in the domain of future work. (2) The authors have done an extensive job of clarifying the differences between these related areas, and as discussed above, other reviewers found the way in which AGP is different to be insightful and ecologically valid. (3) This is potentially the most serious con: as the discussion with Reviewer BHGy brought up, the learned communication protocol may just be a simple mapping between messages and environment interactions. After further discussion in which the authors argued that learning a simple mapping is not a problem---the main question is how to even induce such a mapping in the first place---the reviewer acknowledged that this is not a fatal flaw but that makes the results somewhat less interesting.

In summary, the positive reviews highlighted the interestingness and insightful nature of the questions studied in this paper and have convinced me that this paper will be of interest to the ICLR community as it has provides a new perspective on the problem of agent-agent interaction (particularly for the special case of human-agent interaction). The negative reviews did highlight a few limitations of the paper, but I expect these can be addressed by future work and do not feel they outweigh the interestingness of the problem. In light of this, I recommend acceptance as a poster.

Suggestion for the authors: I found the discussion with Reviewer BHGy to be particularly insightful and helpful in understanding the aims of the paper. I would encourage you to incorporate some of this into the camera-ready version of the paper, and perhaps to lean more heavily on the special case of human-agent interaction as motivation of this work (as also hinted at by Reviewer hMeT).